# Hessian Geometry of Latent Space in Generative Models

**Alexander Lobashev** [1]   **Dmitry Guskov** [2]   **Maria Larchenko** [3]   **Mikhail Tamm** [4]

## Abstract

This paper presents a novel method for analyzing the latent space geometry of generative models, including statistical physics models and diffusion models, by reconstructing the Fisher information metric. The method approximates the posterior distribution of latent variables given generated samples and uses this to learn the log-partition function, which defines the Fisher metric for exponential families. Theoretical convergence guarantees are provided, and the method is validated on the Ising and TASEP models, outperforming existing baselines in reconstructing thermodynamic quantities. Applied to diffusion models, the method reveals a fractal structure of phase transitions in the latent space, characterized by abrupt changes in the Fisher metric. We demonstrate that while geodesic interpolations are approximately linear within individual phases, this linearity breaks down at phase boundaries, where the diffusion model exhibits a divergent Lipschitz constant with respect to the latent space. These findings provide new insights into the complex structure of diffusion model latent spaces and their connection to phenomena like phase transitions. Our source code is available at https://github.com/alobashev/hessian-geometry-of-diffusion-models.

## 1. Introduction

State-of-the-art image generation models often exhibit abrupt changes in image appearance during interpolation, indicating a non-smooth latent space (Liu et al., 2021; Guo et al., 2024). These abrupt transitions have been studied in

[1]Glam AI, San Francisco, USA [2]Artificial Neural Computing Corp., Weston, FL, USA [3]Magicly AI, Dubai, UAE [4]School of Digital Technologies, Tallinn University, Tallinn, Estonia. Correspondence to: Alexander Lobashev <lobashevalexander@gmail.com>, Dmitry Guskov <guskov01dmitry@gmail.com>.

*Proceedings of the 42$^{nd}$ International Conference on Machine Learning*, Vancouver, Canada. PMLR 267, 2025. Copyright 2025 by the author(s).

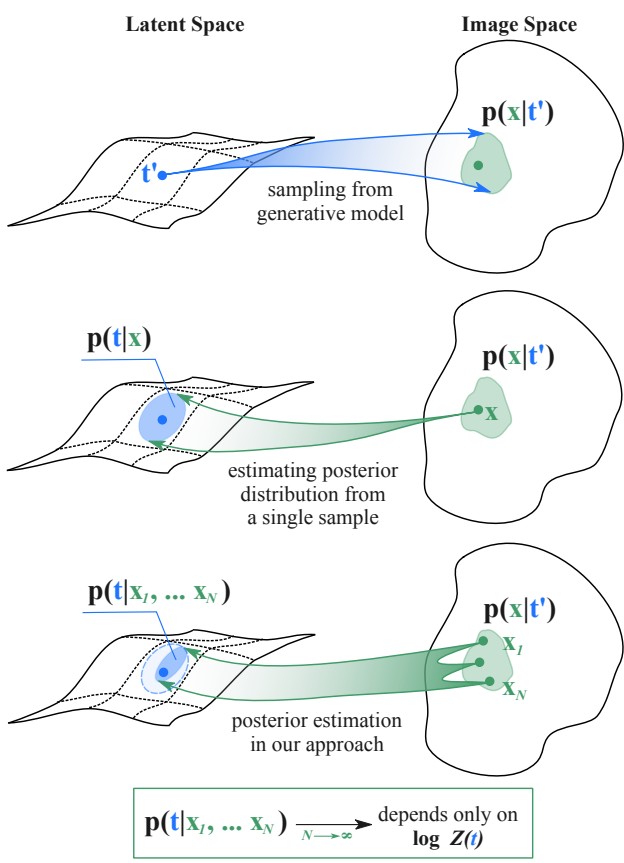

**Latent Space**   **Image Space**

Figure 1: Visualization of Theorem 3.1. Having samples from $p(x|t')$ we can approximate posterior distribution $p(t|x_1, \ldots x_N)$. The partition function $\log Z(t)$ defines a Hessian metric on the latent space. Our Theorem 3.1 guarantees that $\log Z(t)$ in limit depends only on the posterior distributions $p(t|x_1, \ldots, x_N)$ and Theorem 3.2 offers a way to learn $\log Z(t)$ from $p(t|x_1, \ldots, x_N)$.

community from two different perspectives.

**Riemannian geometry of latent space**   Recent work by Park et al. (2023) constructs a latent basis by considering singular vectors of Jacobian in the feature space, deriving a metric on a latent space via pullback from the euclidean metric in the feature space. A similar pullback-based approach from euclidean metric in image space was used by Shao et al. (2018). It was demonstrated that linear interpolation in the latent space closely approximates geodesic interpolation.

Arvanitidis et al. (2017) studied the Riemannian geometry of GAN latent spaces, deriving metrics under the assumption that the stochastic generator's mean and variance functions are twice differentiable. Other notable study (Li et al., 2024) proposes an approach to automatically discover interpretable directions in a latent space. In (Brown et al., 2022) authors verify the union of manifold hypothesis for various image datasets. Additionally, Tosi et al. (2014) treats latent variables as coordinates on a Riemannian manifold, using the Jacobian-derived metric to construct geodesics in latent spaces obtained through nonlinear dimensionality reduction.

**Learning phase transitions in statistical physics**  Machine learning methods have been widely applied to the study of classical and quantum statistical physics models (Van Nieuwenburg et al., 2017; Carrasquilla & Melko, 2017; Rem et al., 2019; Wang et al., 2021; Canabarro et al., 2019). These works primarily focus on determining phase transition boundaries and extracting learned order parameters, which serve to distinguish one phase from another.

In (Walker et al., 2020), it was observed that the principal components of the mean and standard deviation of latent variables in a VAE trained on 2D Ising model configurations were highly correlated with known physical quantities, suggesting that the VAE implicitly extracts sufficient statistics from the data.

We could unify these two approaches by considering a generative model as a statistical physics system and examine it using information geometry methods.

**Our contributions**  This work provides novel method to analyse latent space properties in generative models. It's main contributions are

- For two-parametric systems or two-dimensional sections of the latent space, we propose a method to reconstruct the Fisher metric. We provide a theoretical proof of the method's convergence. The efficiency of the method was tested on exactly solvable statistical physics models: Ising model and TASEP, demonstrating that the reconstructed free energy coincides with the exact solution.

- Using the proposed approach, we analyze the latent spaces of generative models. We introduce the notion of distinct phases within the diffusion latent space and identify boundaries where the recovered Fisher metric exhibits abrupt changes.

- We validate the findings of approximately linear geodesics for interpolation, as discussed in (Shao et al., 2018). However, our work reveals that this result holds only within a single phase. We extend the analysis to phase transitions, showing that the diffusion model exhibits a divergent Lipschitz constant with respect to the latent space at phase boundaries.

## 2. Background

Let $\mathcal{X}$ be a high-dimensional data space and $\mathcal{S}$ a lower-dimensional latent space. We assume the existence of a stochastic generative mapping from $\mathcal{S}$ to $\mathcal{X}$, defined by the conditional probability distribution $p(x|t)$ on $\mathcal{X}$ for each latent vector $t \in \mathcal{S}$.

A generative mapping could be a statistical physics model, such as the Ising model, where $t = (T, H)$ represents the temperature $T$ and external magnetic field $H$, and $x$ represents a spin configuration on a two-dimensional lattice. Alternatively, a generative mapping could be a trained diffusion model using a stochastic sampler. Here, $x$ is a generated image and $t$ is the corresponding latent noise tensor.

### 2.1. Ising Model

We consider the 2D Ising model (Ising, 1925). A microstate $x$ of this model is a set of spin variables $s_i = \pm 1$ defined on a square lattice of size $L \times L$. At equilibrium the probability distribution over the space of microstates is

$$p(x|H,T) = \frac{1}{Z(H,T)} e^{-\frac{1}{T} \sum_{\langle i,j \rangle} s_i s_j - \frac{1}{T} H \sum_i s_i} \quad (1)$$

where $H$ and $T$ are external parameters called magnetic field and temperature. This model is exactly solvable for $H = 0$ (Onsager, 1944; Kac & Ward, 1952; Baxter & Enting, 1978), i.e. $Z(H,T)$ can be analytically found. The model demonstrates a phase transition at $T_{cr} \approx 2.27$ between the high-temperature disordered state, where the distribution is concentrated on microstates where spin variables are on average equal zero and the low-temperature ordered state, where the distribution is concentrated on microstates with non-zero average spin.

### 2.2. Totally Asymmetric Simple Exclusion Process

Totally asymmetric simple exclusion process (TASEP) is a simple model of 1-dimensional transport phenomena (Derrida et al., 1993; Blythe & Evans, 2007; Krapivsky et al., 2010). A microscopic configuration is a set of particles on a 1d lattice. Each particle can move to the site to the right of it with probability $p\,dt$ per time $dt$ provided that it is empty (we put p = 1 without loss of generality). A particular case is open boundary conditions, when a particle is added with probability $\alpha dt$ per time $dt$ to the leftmost site provided that it is empty and removed with probability $\beta dt$ per time $dt$ from the rightmost site provided that it is occupied. For this boundary condition the probability distribution is known exactly:

$$p(x|\alpha,\beta) = \frac{f(x|\alpha,\beta)}{Z(\alpha,\beta)}, \quad (2)$$

where microstate $x$ is a concrete sequence of filled and empty cells. Importantly, the function $f$, which is known exactly does not take the form of the exponential family, Eq.(6). TASEP with free boundaries exhibits a rich phase behavior: For large system sizes three distinct phases - the low-density phase, the high-density phase and the maximal current phase are possible depending on the values of $\alpha, \beta$, and the asymptotic "free energy" equals:

$$F_{\text{TASEP}}(\alpha, \beta) = \begin{cases} \frac{1}{4}, & \alpha > \frac{1}{2}, \ \beta > \frac{1}{2}; \\ \alpha(1-\alpha), & \alpha < \beta, \ \alpha < \frac{1}{2}; \\ \beta(1-\beta), & \beta < \alpha, \ \beta < \frac{1}{2}. \end{cases} \quad (3)$$

### 2.3. Information Geometry

**Fisher metric**  Fisher metric for a distribution $p(x|t)$ is defined as

$$g_F(t) = \int_{\mathcal{X}} p(x|t) \nabla_t \log p(x|t) (\nabla_t \log p(x|t))^T dx \quad (4)$$

The Fisher metric is a Riemannian metric. It equips the space $\mathcal{S}$ of parameters $t$ with the structure of Riemannian manifold $(\mathcal{S}, g_F)$.

**Hessian metric**  A Riemannian metric $g$ is called a Hessian metric if it can be expressed as the Hessian of a convex potential $\phi$ in local coordinates:

$$g = \sum_{i,j=1}^{N} \frac{\partial^2 \phi(x)}{\partial x^i \partial x^j} dx^i dx^j. \quad (5)$$

**Exponential Family**  The exponential family consists of distributions of the form

$$p(x|t) = e^{\langle f(x), t \rangle - \log Z(t)}, \quad (6)$$

where the partition function $Z(t)$ is given by

$$Z(t) = \int_{\mathcal{X}} e^{\langle f(x), t \rangle} dx. \quad (7)$$

The function $f(x)$ called unnormalized density in machine learning and Hamiltonian in statistical physics. It is generally unknown, making the direct integration of Eq.7 impossible.

A key property of exponential families is that their Fisher metric is always Hessian, equaling the Hessian of the log-partition function:

$$g_F(t) = \sum_{i,j=1}^{N} \frac{\partial^2 \log Z(t)}{\partial t^i \partial t^j} dt^i dt^j = \nabla^2 \log Z(t) \quad (8)$$

**Hessianizability**  A natural question arises: When does a Riemannian metric admit a Hessian structure? The Bryant–Amari–Armstrong theorem (Bryant, 2013; Amari & Armstrong, 2014; Bryant, 2024) states that this is always locally true for 2D analytic manifolds, later extended to smooth cases (Bryant, 2024):

**Theorem 2.1.** *(Bryant–Amari–Armstrong) Any analytic Riemannian metric on a 2-dimensional manifold locally admits a Hessian representation.*

While Theorems 3.1 and 3.2 presented in the next section apply to arbitrary dimensions of data $\mathcal{X}$ and latent space $\mathcal{S}$, they require a special (exponential) form of the data distribution. Theorem 2.1 allows us to analyze 2D subspace of latent spaces in GANs, diffusion models, and other non-exponential generative models. Notably, the approach proposed below is theoretically justified for *any* generative model.

## 3. Method

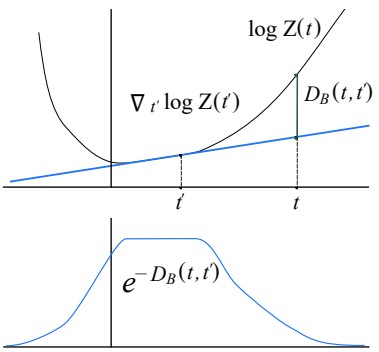

Figure 2: Visualizing Bregman divergence. **(Top)** A convex function $\log Z(t)$ (black curve) and its tangent line (blue) at a point $t'$. The Bregman divergence $D_B(t, t')$ (vertical green segment) measures the difference between $\log Z(t)$ and the linear approximation at $t'$. **(Bottom)** The exponential distribution $e^{-D_B(t,t')}$ (blue surface) approximates the posterior distribution on parameters $p(t|x_1, \ldots, x_N)^{\frac{1}{N}}$, where $x_1, \ldots, x_N \sim p(x|t')$, illustrating Theorem 3.1.

This section describes a method for approximating the Fisher metric on the latent space of a generative model, $p(x|t)$, enabling the computation of geodesics for smoother interpolation and the identification of phase transitions.

Given a generative model $p(x|t)$ we proceed in two steps (1) approximating the posterior distribution $p(t|x)$, and (2) estimating the log partition function $\log Z(t)$ by training a network to simulate $p(t|x)$. Theorems 3.1 and 3.2 justify this approach for exponential families.

**Theorem 3.1.** *Let $\mathcal{X}$ be a space of data samples $x \in \mathcal{X}$, and $S \subset \mathbb{R}^n$ be a compact domain with the continuous prior distribution $p(t)$ supported on $S$. Suppose the conditional distribution of data samples given parameter $t$ is an*

*exponential family*

$$p(x|t) = e^{\langle t, f(x)\rangle - \log Z(t)}, \tag{9}$$

*where*

$$Z(t) = \int_{\mathcal{X}} e^{\langle t, f(x)\rangle} dx \tag{10}$$

*converges for all $t \in S$. Let $x_1, \ldots, x_N \sim p(x|t')$. Then, as $N \to \infty$ the posterior distribution satisfies:*

$$\lim_{N \to \infty} (p(t|x_1, \ldots, x_N))^{1/N} \stackrel{a.s.}{=} e^{-D_{\log Z(t)}(t,t')} \tag{11}$$

*where $D_{\log Z(t)}(t,t')$ is the Bregman divergence between exponential family distributions*

$$\begin{aligned} D_{\log Z(t)}(t,t') = \\ = \log Z(t) - \log Z(t') - \langle \nabla_{t'} \log Z(t'), t - t' \rangle \end{aligned} \tag{12}$$

For exponential families, $D_{\log Z(t)}(t,t')$ coincides with the Kullback-Leibler divergence $D_{\mathrm{KL}}(p(x|t')\|p(x|t))$. Theorem 3.1 thus implies that as more data samples are observed the posterior concentrates on parameters minimizing KL divergence with the true $t'$. For intuition, see Fig.2.

**Theorem 3.2.** *Suppose that the following integral converges to zero*

$$\int_S \int_S \left| e^{-D_{\log Z_1(t)}(t,t')} - e^{-D_{\log Z_2(t)}(t,t')} \right|^2 dt\, dt' \to 0, \tag{13}$$

*where $D_{\log Z_1(t)}(t,t')$ is the Bregman divergence. Then the Hessian of $\log Z_1(t)$ converges to the Hessian $\log Z_2(t)$ uniformly in $t$*

$$||\nabla^2 \log Z_1(t) - \nabla^2 \log Z_2(t)|| \to 0, \tag{14}$$

*where $|| \cdot ||$ denotes the $L^2$ norm.*

In other words, Eq. 13 is MSE loss function for training $\log Z(t)$. Strictly speaking, it obtains the partition function only up to an affine transformation $\log Z(t) \sim \log Z(t) + \langle c, t \rangle + b$. However, our goal is to examine a metric $g$ which is a Hessian $g(t) = \nabla^2 \log Z(t)$ and therefore it does not depend of affine term. Unfortunately, training with MSE loss suffers from vanishing gradients during initial optimization stages. Loss selection for the experiment is covered in Section 3.2.

For simplicity, we assume a uniform parameter distribution $p(t)$ over $S$:

$$p(t) = \begin{cases} \frac{1}{\mathrm{Vol}(S)}, & t \in S, \\ 0, & \text{otherwise.} \end{cases} \tag{15}$$

This choice avoids bias toward specific regions of $S$, ensuring equitable exploration of the latent space.

## 3.1. Approximation of the Posterior

The first step is illustrated in Fig. 1. During this step we obtain an approximation of the posterior distribution from a set of samples $p(t|x_1, \ldots, x_N)$.

**Training a Mapping** One could deal with this task by directly training a mapping model, that takes as input $x_1, \ldots, x_N$ sampled from $p(x|t')$ and returns a normalized probability distribution on a compact domain $S$. This approach is most suitable when the samples have stochastic nature and no feature extractor can be utilized.

This is the case for Ising and TASEP statistical systems, where a sample $x_i$ is a microstate, i.e. it is one of the many indistinguishable realizations of the system with the given parameters $t$. Importantly, such samples can be pixel-wise uncorrelated. Therefore in this case we use $U^2$-Net (Qin et al., 2020) trained via maximizing likelihood of the true parameters $t'$. For the training details please refer to the Appendix B.3.

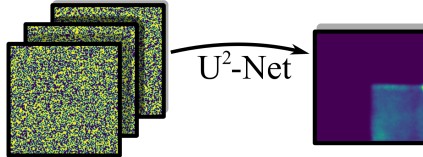

**Using a Feature Extractor** In the image domain, one could use a pre-trained feature extractor. Note that from Theorem 3.1, it follows that the posterior distribution behaves as $\sim \exp(-ND_{\mathrm{KL}}(p_1 \| p_2))$. Having a pre-trained feature extractor $\mathcal{E}$, we can construct an approximation of $D_{\mathrm{KL}}$ in terms of a distance between feature vectors:

$$\begin{aligned} D_{\mathrm{KL}}(p(x|t_1) \| p(x|t_2)) &\approx d(\mathcal{E}(x_1), \mathcal{E}(x_2)), \\ x_1 &\sim p(x|t_1), \\ x_2 &\sim p(x|t_2). \end{aligned} \tag{16}$$

Indeed, if the feature extractor $\mathcal{E}$ acts as an approximate sufficient statistic, then

$$D_{\mathrm{KL}}(p(x|t_1) \| p(x|t_2)) = D_{\mathrm{KL}}(p(\mathcal{E}(x)|t_1) \| p(\mathcal{E}(x)|t_2)). \tag{17}$$

If the distribution of features $p(\mathcal{E}(x)|t_i)$ under $t_i$ is approximately normal $\mathcal{N}(\mu_i, I)$, then the KL divergence simplifies (up to an additive constant) to the squared difference between the feature means:

$$D_{\mathrm{KL}}(p(\mathcal{E}(x)|t_1) \| p(\mathcal{E}(x)|t_2)) = \frac{1}{2}\|\mu_1 - \mu_2\|^2. \tag{18}$$

Then the posterior could be approximated as

$$\begin{aligned} p(t|x_1, \ldots, x_N) &\approx e^{-\frac{N}{2}\|\mathcal{E}(x) - \mathcal{E}(x')\|^2}, \\ x &\sim p(x|t), \; x' \sim p(x|t'). \end{aligned} \tag{19}$$

Below in the Experiments section we take CLIP as a feature extractor $\mathcal{E}$ and produce the approximation of posterior distribution based on distances between CLIP images embeddings.

### 3.2. Approximation of the Fisher Metric

The loss function in Equation 13, while theoretically ensuring Hessian convergence as stated in Theorem 3.2, suffers from vanishing gradients during the initial optimization stages (see lemma A.6 from the Appendix section). To address this problem, we normalize $e^{-ND_{\log Z(t)}(t,t')}$ to obtain a probability distribution. Then we compare it with the posterior distribution $p(t \mid x_1, \ldots, x_N)$ using Jensen-Shannon divergence (JSD), which is a proper metric between two probability distributions.

**Lemma 3.3.** *Let $Z : S \to (0, \infty)$ with $S \subset \mathbb{R}^n$ compact domain and $\log Z$ convex. Then for any fixed $t \in S$:*

$$
\frac{\exp\left(-D_{\log Z}(t, t')\right)}{\int_S \exp\left(-D_{\log Z}(s, t')\right) ds} =
$$
$$
= \frac{\exp\left(-\langle t, \nabla_{t'} \log Z(t')\rangle + \log Z(t)\right)}{\int_S \exp\left(-\langle s, \nabla_{t'} \log Z(t')\rangle + \log Z(s)\right) ds}, \quad (20)
$$

*where $D_{\log Z}(t, t')$ is the Bregman divergence.*

Define a normalized distribution which depends only on the log-partition function following Lemma 3.3

$$
p_{\log Z}(t|t') = \frac{\exp\left(-\langle t, \nabla_{t'} \log Z(t')\rangle + \log Z(t)\right)}{\int_S \exp\left(-\langle s, \nabla_{t'} \log Z(t')\rangle + \log Z(s)\right) ds} \quad (21)
$$

Now given the posterior $p(t|x_1, \ldots, x_N)$, which approximation was discussed in the previous section, we could train the log-partition function $\log Z_\theta(t)$ by minimizing the loss

$$
\mathcal{L}_1(\theta) = \int_{\mathcal{S}} D_{\text{JS}}\left(p(t|x_1, \ldots, x_N), p_{\log Z_\theta}(t|t')\right) dt', \quad (22)
$$

where $x_1, \ldots, x_N \sim p(x|t')$. The Jensen-Shannon divergence for distributions $P$ and $Q$ is defined as

$$
D_{\text{JS}}(P, Q) = \frac{1}{2}\left[D_{KL}(P\|\frac{P+Q}{2}) + D_{KL}(Q\|\frac{P+Q}{2})\right] \quad (23)
$$

The resulting approximation of the Fisher metric is

$$
g_F(t) = \nabla^2 \log Z_{\theta^*}(t), \ \theta^* = \underset{\theta}{\arg\min} \mathcal{L}(\theta) \quad (24)
$$

We model $\log Z_\theta(t)$ by MLP with 5 hidden layers, hidden size of 512 with ReLU activation. We do not require the MLP to be convex as it converges to the convex function during the training.

### 3.3. Geodesic Approximation

After obtaining the Fisher metric, $g_F$, we unlock the ability to explore the intrinsic geometry of our statistical model space. Geodesics, in this context, represent the shortest paths between two probability distributions *within* this space. They are analogous to straight lines on a flat surface, but in a potentially curved space dictated by the Fisher metric. To find these geodesics, we aim to minimize the curve length, $L[\gamma(t)]$, for a smooth curve $\gamma(t)$ parameterized from $t = 0$ to $t = 1$ and lying within our statistical model space. This curve length is calculated using the Fisher metric as:

$$
L[\gamma(t)] = \int_0^1 \sqrt{\dot{\gamma}(t)^T g_f(\gamma(t))\dot{\gamma}(t)} dt \quad (25)
$$

Here, $\gamma(t)$ represents a path in the parameter space, and $\dot{\gamma}(t) = \frac{d\gamma(t)}{dt}$ is its tangent vector.

We split $\gamma(t)$ into discrete points $\{\gamma_0, \gamma_1, \ldots, \gamma_N\}$, where $\gamma_0$ and $\gamma_N$ are the starting and ending points of our desired geodesic. The continuous integral is then represented by a discrete sum of distances between consecutive points, using the Fisher metric to measure these distances, optimizing intermediate points via Adam to minimize $L[\gamma]$ (see (Shao et al., 2018) for details).

## 4. Experiments and Results

To support our theoretical reasoning, in this section validate our method on the exactly solvable statistical models, namely Ising and TASEP, and compare our estimation of $\log Z(t)$ with a ground truth. Then, we evaluate our method on two-dimensional slices of diffusion models, comparing the path length and curvature of the learned trajectories with those produced by other methods.

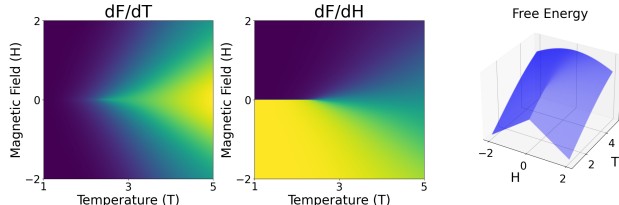

Figure 3: 2D Ising model. Partial derivative of the reconstructed free energy with respect to temperature $\frac{\partial F_{\text{rec}}(T,H)}{\partial T}$, Partial derivative of the reconstructed free energy with respect to magnetic field $\frac{\partial F_{\text{rec}}(T,H)}{\partial H}$ and reconstructed free energy. See Appendix B for dataset and training details.

### 4.1. Exactly Solvable Statistical Models

We compare against two baselines: (1) *posterior-mean-as-statistics* (Mean-as-Stat) approach (Fearnhead & Prangle,

Ours      Ground truth

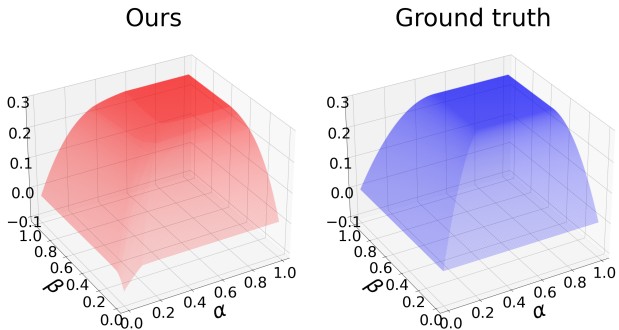

Figure 4: TASEP. Left: reconstructed free energy (red) compared to the exact solution (blue). Our method achieves near-exact agreement except near domain boundaries (Appendix B.1)

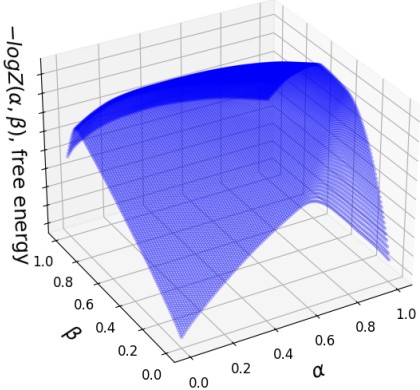

Figure 5: Free energy of diffusion model reconstructed with CLIP distance.

2012; Jiang et al., 2017), and (2) *PCA-VAE*, a dimensionality reduction approach following (Walker et al., 2020) (see Appendix B.1 for details of baselines evaluation).

Table 1 compares the performance of our method ("Convex") against two baselines on 2D Ising and TASEP models. We report the root mean squared error (RMSE) for the reconstructed free energy ($F$) and its partial derivatives with respect to the model parameters ($\frac{dF}{dT}$, $\frac{dF}{dH}$ for Ising; $\frac{dF}{d\alpha}$, $\frac{dF}{d\beta}$ for TASEP). Accurate reconstruction of partial derivatives is critical for identifying phase transitions, where these derivatives diverge. Our method achieves lower RMSE across all metrics in reconstructing thermodynamic quantities.

Fig.3 and Fig.4 demonstrate the results of free energy $F \sim \log Z(t)$ reconstruction for both systems.

### 4.2. Two-dimensional Slices of a Diffusion Model

The experiments with diffusion are based on StableDiffusion 1.5 (Dreamshaper8) (Lykon, 2023) with DDIM scheduler

Table 1: Performance results (RMSE) for ISING and TASEP models. F stands for Free Energy.

| **ISING** | F RMSE | $\frac{dF}{dT}$ RMSE | $\frac{dF}{dH}$ RMSE |
|---|---|---|---|
| Convex (Ours) | **0.0883** $\pm$ **0.0006** | **0.1106** $\pm$ **0.0002** | **0.1237** $\pm$ **0.0016** |
| Mean-as-Stat | 0.0981 $\pm$ 0.0010 | 0.4766 $\pm$ 0.0023 | 1.0936 $\pm$ 0.0033 |
| PCA-VAE | 0.1669 $\pm$ 0.0018 | 0.7428 $\pm$ 0.0025 | 0.7988 $\pm$ 0.0022 |
| **TASEP** | F RMSE | $\frac{dF}{d\alpha}$ RMSE | $\frac{dF}{d\beta}$ RMSE |
| Convex (Ours) | **0.0112** $\pm$ **0.00008** | **0.1165** $\pm$ **0.0025** | **0.1135** $\pm$ **0.0017** |
| Mean-as-Stat | 0.0529 $\pm$ 0.0005 | 0.3832 $\pm$ 0.0038 | 0.3833 $\pm$ 0.0031 |
| PCA-VAE | 0.0524 $\pm$ 0.0006 | 0.3837 $\pm$ 0.0038 | 0.3872 $\pm$ 0.0022 |

(Song et al., 2020). For our generation we use 50 inference steps, classifier free guidance scale set to 5. Prompt is chosen as "High quality picture, 4k, detailed" and negative prompt "blurry, ugly, stock photo".

To build a 2 dimensional latent space section of the diffusion model we generate 3 random initial latents $z_0, z_1, z_2$. We use interpolation between latent representations:

$$\mathbf{z} = z_0 + \alpha(z_1 - z_0) + \beta(z_2 - z_0),$$

where $\alpha$ and $\beta$ are uniformly sampled from $U[0, 1]$. In this setup we take three different triplets of initial latents and generate 60000 images for each with a fixed prompt and initial latent uniformly sampled from the interpolation grid. To prevent our interpolation to fall off the Gaussian hypersphere in all our experiments we employed a normalization of latent vector z.

At first we limit ourselves to the case of deterministic generation process, setting DDIM parameter $\eta = 0$.

**Using a CLIP Feature Extractor** For the given dataset we then compute CLIP distances to approximate prior distribution as discussed in Sec. 3.1. Following our method we train model to reconstruct $\log Z(t) = \log Z(\alpha, \beta)$ presented in Fig 5. The retrieved $\log Z(t)$ is not smooth and has abrupt changes in derivatives shown on Fig.7(C), reflecting phase transitions in image space. To evaluate our emergent metric, we analyze geodesic paths. Our results demonstrate that geodesics are approximately linear within a single phase but exhibit nonlinear deviations at phase boundaries, Fig. 7(B). The Fisher metric enables the construction of smoother interpolation paths (geodesics) compared to linear interpolation, Fig. 7(A)

We then investigate images near the phases boundaries. Unlike classical statistical physics models with continuous

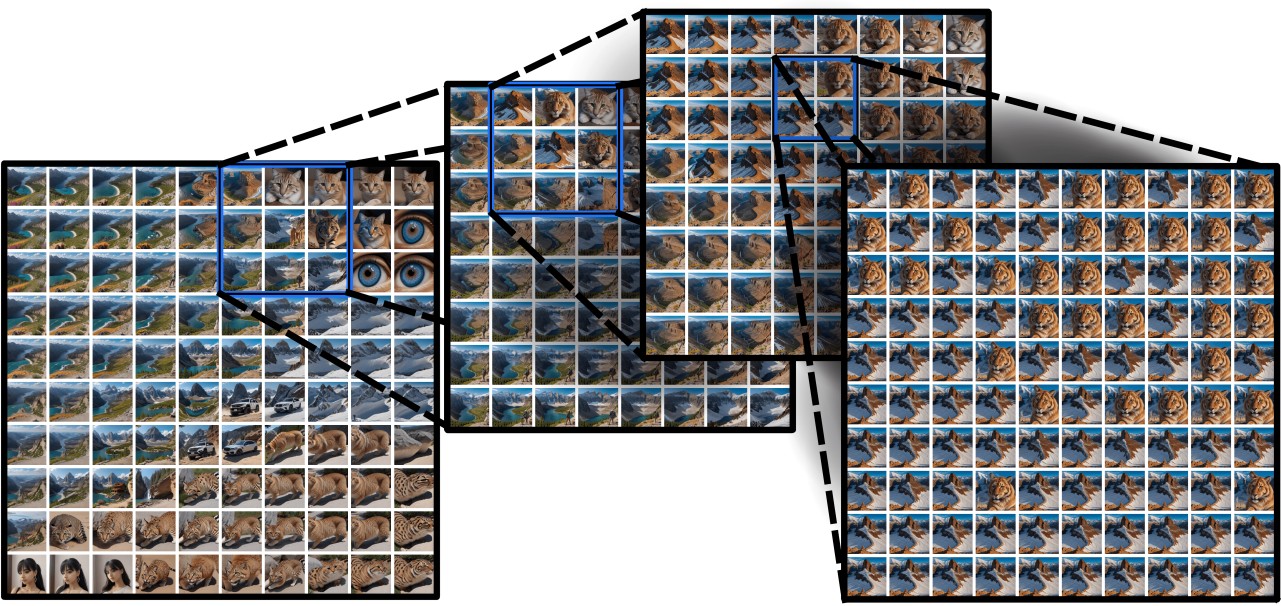

Figure 6: The fractal structure of phase boundary in the interpolation landscape of diffusion model. The last plot represent the parameter variations $10^{-5}$ between neighboring images. The small changes in latent space cause switching between a mountain and a lion.

phase boundaries, the latent diffusion model exhibits phase diagrams with fractal boundaries, illustrated by Fig.6.

Crucially, we observe that near these boundaries, samples from neighboring phases blend together, leading to a phenomenon where phases **permeate** one another. Repeated magnification of the boundary shown in Fig.6 reveals self-similar patterns across scales, starting from variation scale $10^{-5}$ until reaching float16 accuracy of $10^{-8}$. In other words, near the phase boundaries the diffusion output is highly sensitive to small changes in latent space.

This behaviour is commonly characterized by the Lipschitz constant. The work by Yang et al. (Yang et al., 2023) discusses the Lipshitz constant for diffusion models with respect to time variable. However, to our best knowledge, the observation on divergence of Lipshitz constant with respect to latent space is new.

The Fisher metric is no longer analytic or smooth there, meaning the Brayant-Amari-Armstrong theorem does not apply, and the metric cannot be expressed as the Hessian of a convex function.

**Using a Unet Mapping** We investigate our method on diffusion model with $U^2$-Net and observe a distinct outcome. Unet doesn't take into account semantic information, solely relying on pixel space representation. Therefore it is able to predict the exact parameters $(\alpha, \beta)$ of image generation. From this perspective the generation is completely deterministic and every image is distinguishable. Thus, pos-

terior distribution exactly recovers the target one resulting in smooth $\log Z(\alpha, \beta)$.

Following this result, we evaluate our method on generation with DDIM $\eta = 0.1$. We observe that adding the noise simply results in blurring the posterior distribution prediction. In the case of $U^2$-Net it doesn't change the learned geometry, following the argument above. However, in the case of CLIP this results in smoothed free energy landscape and thus doesn't show sharp boundaries as in noiseless case.

**Baselines and Metrics** To evaluate our method, we compare it against the approaches of Wang et al. (2021) and Shao et al. (2018), which represent related work in deterministic generative models. Below, we clarify the key differences and present quantitative results from these comparisons.

Shao et al. (2018) study VAEs and compute geodesic curves based on the pullback metric induced by the Euclidean metric in pixel space. Wang et al. (2021) consider GAN models and define a metric on the latent space via the pullback of the $\ell_2$ distance in the feature space of VGG-19 (LPIPS distance). In both cases, the generative models are deterministic: each latent $Z$ produces only a single image $X$.

The key distinction of our work lies in its consideration of a broader class of models, specifically, models with stochastic generation, where a single latent $Z$ corresponds to a distribution in the image space $p(X|Z)$. Diffusion models with stochastic sampling (and all statistical physics models) cannot be addressed within the formalism of Shao et al. (2018)

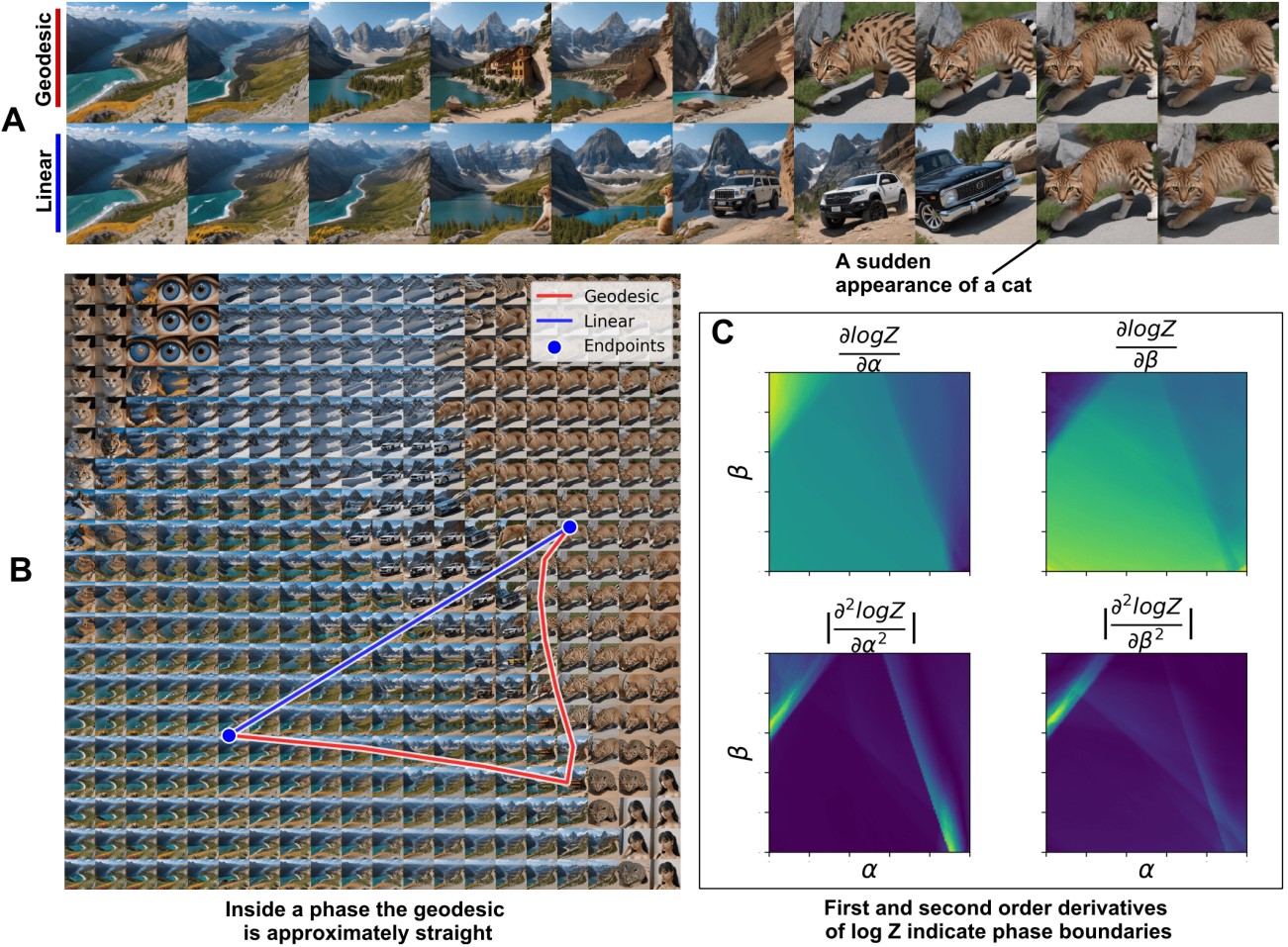

Figure 7: (A) Geodesic (ours) and linear interpolation between images. Note that our interpolation variant is more perceptually smooth. (B) Illustration of 60 000 generated by StableDiffusion images with geodesic plots (C) First and second derivatives of the log-partition function of diffusion model. Note, that second derivatives are diagonal components of the Fisher metric.

or Wang et al. (2021).

We compare our algorithm with these baselines in the deterministic sampling regime. We obtain the pullback metric from the Euclidean metric in the space of CLIP embeddings. The Jacobian is estimated via finite differences. We use three evaluation scores: CLIP, pixel, and Perceptual Path Length (PPL) (Karras et al., 2019), which measure the average path length in CLIP embedding space, pixel space, and the feature space of VGG-19, respectively. These scores are computed as the cumulative $\ell_2$ distance between consecutive feature vectors (or images for pixel space), then averaged across multiple trajectories (Table 2).

We observe that our interpolation is on par with baselines in the case of deterministic sampling. We additionally compute the curvature of each trajectory as the mean angular change per unit length between consecutive path segments. Our

analysis reveals that trajectories constructed using the Wang metric show significantly higher curvature and frequent turning compared to our method. We attribute this behavior to the finite differences, which introduce high-frequency noise in the metric components. In the case of diffusion, the Jacobian is hard to obtain via backpropagation, as suggested in Shao et al. (2018), due to high computational cost. During our evaluation we observed that the phase boundaries remained stable across all tested feature extractors. Therefore, we conclude that the algorithm consistently captures the same phase structure.

**Underlying mechanism for phase transition** We believe the observed phase transition behavior is connected to the geometry of the image space. This idea aligns with findings from the Brown et al. (2022), which shows that natural images lie on a union of disjoint low-dimensional manifolds with varying dimensions.

Table 2: Comparison of trajectory lengths and curvature in deterministic sampling regime.

| Metric | Geodesic (Ours) | Linear | Geodesic (Wang/Shao) |
|---|---|---|---|
| CLIP Length | **72.3 ± 4.00** | 73.6 ± 3.54 | 73.6 ± 4.37 |
| Pixel Length | $2.77 \times 10^6 \pm 2.38 \times 10^4$ | $2.76 \times 10^6 \pm 2.77 \times 10^4$ | $\mathbf{2.74 \times 10^6 \pm 3.53 \times 10^4}$ |
| Perceptual Path Length | **3.12 ± 0.16** | 3.17 ± 0.23 | 3.19 ± 0.21 |
| Mean Curvature | **0.367 ± 0.691** | 0.00 ± 0.00 | 1.33 ± 0.53 |

In diffusion models, generation begins from a high-dimensional Gaussian latent distribution. The reverse ODE process maps this distribution onto disjoint, lower-dimensional manifolds corresponding to distinct image modes. Such a transformation—from a unimodal latent space to a multimodal data space with disjoint supports—may result in a diverging Lyapunov exponent or, equivalently, a diverging Lipschitz constant in the generative mapping, indicating phase transitions.

This phenomenon can be illustrated by the following proposition, which simulates a lower-dimensional data manifold with disjoint supports.

**Proposition 4.1.** *Suppose that the (target) data distribution is a bimodal mixture of two Gaussians, each with variance $\sigma^2$:*

$$p_0(x) = \frac{1}{2}\mathcal{N}(x \mid -1, \sigma^2) + \frac{1}{2}\mathcal{N}(x \mid 1, \sigma^2). \quad (26)$$

*The latent (source) distribution is the standard normal $\mathcal{N}(x \mid 0, 1)$. Consider the variance-preserving SDE*

$$dX_t = -\frac{1}{2}\beta X_t \, dt + \sqrt{\beta} \, dW_t. \quad (27)$$

*Then the Lyapunov exponent of the corresponding reverse-time ODE at $x = 0$ has the following form:*

$$\lambda = \frac{\beta}{2}\left(1 + \frac{1 - \sigma^2}{\sigma^4}\right), \quad (28)$$

*and it diverges to infinity as $\sigma \to 0$. In this case, the point $x = 0$ can be interpreted as a phase transition boundary.*

## 5. Discussion

In this work we develop a method to reconstruct the Fisher metric for any exponential family distribution. This is typical for statistical physics systems. Our method is based on reconstructing the log-partition function from posterior distributions on the parameter space. We provide theoretical grounding and validate it through experiments by comparing our reconstruction with exact solutions, outperforming mean-as-sufficient-statistics and PCA-VAE baselines. Importantly, we do not assume prior knowledge of the unnormalized density or Hamiltonian, enabling our method to

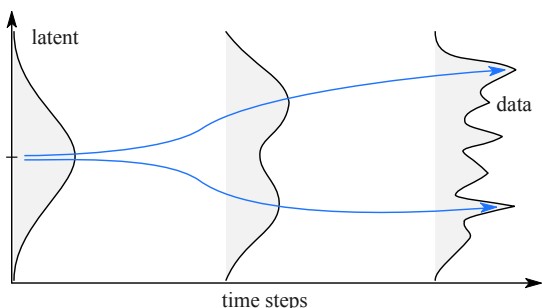

Figure 8: Illustration of Proposition 4.1

operate in more general settings than MCMC-based techniques like importance sampling.

Bryant-Amari-Armstrong theorem allows us to go beyond exponential family distributions and apply our method to study two-dimensional sections of latent space of arbitrary generative models.

We extend notion of phase transitions known in statistical physics, where small changes in parameters lead to significant shifts in system output, to describe abrupt changes in generative models. While recent studies (Sclocchi et al., 2025; Biroli et al., 2024) have explored phase transitions in diffusion models with respect to time-step, our work addresses a distinct problem. We treat the latent space as a parameter space and generated images as microstates. Our reconstruction of the Fisher metric for a two-dimensional section of a diffusion model's latent space indicates the presence of distinct phases. We observe that within a phase, geodesic interpolations between images are approximately linear, consistent with the work of Shao et al. (Shao et al., 2018). However, our results reveal that this assumption breaks down at phase boundaries, where the geometry becomes highly nonlinear. Specifically, we observe that diffusion models exhibit fractal boundaries between phases. We argue that near these boundaries diffusion has divergent Lipschitz constant. It complements results of Yang et al. (Yang et al., 2023) showing that diffusion has divergent Lipschitz constant in time variable.

## Impact Statement

Our work contributes into the fundamental understanding of generative model latent spaces, offering mathematically grounded tools for detecting phase transitions and constructing smoother geodesic interpolations. While generative models in general raise concerns about misuse, our theoretical focus does not directly enable such risks. Nonetheless, we acknowledge that better exploration of latent space could, in principle, facilitate adversarial prompt crafting or model probing. We expect our contributions to support better understanding of generative models and stimulate cross-disciplinary dialogue between machine learning and information geometry communities.

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

## A. Proof of theoretical results

**Theorem A.1.** *Let $\mathcal{X}$ be a space of data samples $x \in \mathcal{X}$, and $S \subset \mathbb{R}^n$ be a compact domain with the continuous prior distribution $p(t)$ supported on $S$. Suppose the conditional distribution of data samples given parameter $t$ is an exponential family*

$$p(x|t) = e^{\langle t, f(x) \rangle - \log Z(t)}, \tag{29}$$

*where*

$$Z(t) = \int_{\mathcal{X}} e^{\langle t, f(x) \rangle} dx \tag{30}$$

*converges for all $t \in S$. Let $x_1, \ldots, x_N \sim p(x|t')$. Then, as $N \to \infty$ the posterior distribution satisfies:*

$$\lim_{N \to \infty} (p(t|x_1, \ldots, x_N))^{1/N} \stackrel{a.s.}{=} e^{-D_{\log Z(t)}(t,t')} = e^{-D_{KL}(p(x|t') \| p(x|t))}. \tag{31}$$

*where $D_{\log Z(t)}(t, t')$ is the Bregman divergence between exponential family distributions*

$$D_{\log Z(t)}(t, t') = \log Z(t) - \log Z(t') - \langle \nabla_{t'} \log Z(t'), t - t' \rangle \tag{32}$$

*Proof.* We begin by applying Bayes' theorem to derive the posterior distribution. Then we use properties of exponential families and the law of large numbers to obtain the desired limit.

By Bayes' theorem:

$$p(t|x_1, \ldots, x_N) = \frac{p(x_1, \ldots, x_N|t)p(t)}{\int_S p(x_1, \ldots, x_N|s)p(s)ds}. \tag{33}$$

Since the data samples $x_1, \ldots, x_N$ are i.i.d. given $t$, we have:

$$
\begin{aligned}
p(x_1, \ldots, x_N|t) &= \prod_{i=1}^{N} p(x_i|t) = \prod_{i=1}^{N} e^{\langle t, f(x_i) \rangle - \log Z(t)} \\
&= e^{\sum_{i=1}^{N} \langle t, f(x_i) \rangle - N \log Z(t)}.
\end{aligned}
\tag{34}
$$

Substituting into the posterior, we find:

$$p(t|x_1, \ldots, x_N)^{1/N} = \frac{e^{\langle t, \frac{1}{N} \sum_{i=1}^{N} f(x_i) \rangle - \log Z(t) + \frac{1}{N} \log p(t)}}{\left( \int_S e^{N \left[ \langle s, \frac{1}{N} \sum_{i=1}^{N} f(x_i) \rangle - \log Z(s) + \frac{1}{N} \log p(s) \right]} ds \right)^{1/N}}. \tag{35}$$

Now, consider the limit:

$$\lim_{N \to \infty} \frac{1}{N} \sum_{i=1}^{N} f(x_i). \tag{36}$$

By Kolmogorov's strong law of large numbers, this converges almost surely to the expectation:

$$\frac{1}{N} \sum_{i=1}^{N} f(x_i) \xrightarrow{a.s.} \int f(x) p(x|t') dx = \nabla_{t'} \log Z(t'), \tag{37}$$

where the last equality holds because $p(x|t)$ is an exponential family distribution.

In other words:

$$p\left( \left\{ \omega \in \Omega : \lim_{N \to \infty} \frac{1}{N} \sum_{i=1}^{N} f(x_i(\omega)) = \nabla_{t'} \log Z(t') \right\} \middle| t' \right) = 1. \tag{38}$$

By the continuous mapping theorem: if $X_1, X_2, \ldots$ is a sequence of random variables and $h : \mathbb{R} \to \mathbb{R}$ is a continuous function, then:

$$X_n \xrightarrow{a.s.} X \implies h(X_n) \xrightarrow{a.s.} h(X). \tag{39}$$

Let $\psi(s,x) = e^{\langle s,x\rangle - \log Z(s) + \log p(s)}$. Since the function $\psi$ is continuous in $x$, and we have shown that $\frac{1}{N}\sum_{i=1}^{N} f(x_i) \xrightarrow{\text{a.s.}} \nabla_{t'} \log Z(t')$ as $N \to \infty$, the continuous mapping theorem implies:

$$e^{\langle t, \frac{1}{N}\sum_{i=1}^{N} f(x_i)\rangle - \log Z(t) + \frac{1}{N}\log p(t)} \xrightarrow{\text{a.s.}} e^{\langle t, \nabla_{t'} \log Z(t')\rangle - \log Z(t)}, \tag{40}$$

where $p(t)$ is a continuous function on a compact domain, and therefore bounded, which implies $\frac{1}{N}\log p(t) \to 0$ as $N \to \infty$.

Now, let's consider the term in the denominator of the posterior :

$$\left(\int_S e^{N\left[\langle s, \frac{1}{N}\sum_{i=1}^{N} f(x_i)\rangle - \log Z(s) + \frac{1}{N}\log p(s)\right]} ds\right)^{1/N}. \tag{41}$$

We can rewrite the integral part using Lemma A.2. This lemma applies because, according to the strong law of large numbers, the average $\frac{1}{N}\sum_{i=1}^{N} f(x_i)$ converges almost surely to a finite value. To utilize the lemma, we define the function:

$$\phi(s, N, \omega) = \langle s, \frac{1}{N}\sum_{i=1}^{N} f(x_i(\omega))\rangle - \log Z(s) = \psi\left(s, \frac{1}{N}\sum_{i=1}^{N} f(x_i(\omega))\right). \tag{42}$$

We will ignore the term $\frac{1}{N}\log p(s)$ as it converges to 0 as N goes to infinity and wont affect the value of the integral. Then by Lemma A.2, we can state that the following limit holds almost surely:

$$\lim_{N \to \infty} \frac{1}{N}\log \int_S e^{N\phi(s,N,\omega)} ds \stackrel{\text{a.s.}}{=} \max_{s \in S} \psi(s, \nabla_{t'} \log Z(t')). \tag{43}$$

Using this result, we have

$$p(t|x_1, \ldots, x_N)^{1/N} = \frac{e^{\langle t, \frac{1}{N}\sum_{i=1}^{N} f(x_i)\rangle - \log Z(t) + \frac{1}{N}\log p(t)}}{\left(\int_S e^{\langle s, \frac{1}{N}\sum_{i=1}^{N} f(x_i)\rangle - \log Z(s) + \frac{1}{N}\log p(s)} ds\right)^{1/N}} \xrightarrow{\text{a.s.}} \frac{e^{\langle t, \nabla_{t'} \log Z(t')\rangle - \log Z(t)}}{e^{\max_{s \in S} \psi(s, \nabla_{t'} \log Z(t'))}}. \tag{44}$$

Now, observe that we have:

$$\max_{s \in S} \psi(s, \nabla_{t'} \log Z(t')) = \max_{s \in S}\left[\langle s, \nabla_{t'} \log Z(t')\rangle - \log Z(s)\right]. \tag{45}$$

Since the function is concave, we can state that the maximum is achieved when $s = t'$, which allows us to obtain

$$\max_{s \in S} \psi(s, \nabla_{t'} \log Z(t')) = \langle t', \nabla_{t'} \log Z(t')\rangle - \log Z(t') \tag{46}$$

Substituting back into the limit, we arrive at the expression for the Bregman divergence:

$$\lim_{N \to \infty} p(t|x_1, \ldots, x_N)^{1/N} \stackrel{\text{a.s.}}{=} \frac{e^{\langle t, \nabla_{t'} \log Z(t')\rangle - \log Z(t)}}{e^{\langle t', \nabla_{t'} \log Z(t')\rangle - \log Z(t')}} = e^{-D_B(t',t)}. \tag{47}$$

To complete the proof, we need to show that the KL divergence $D_{\text{KL}}(p(x|t')\|p(x|t))$ in the theorem statement is equivalent to the Bregman divergence $D_B(t,t')$ in our result. We compute the KL divergence as follows:

$$\begin{aligned}
D_{\text{KL}}(p(x|t')\|p(x|t)) &= \int p(x|t') \log \frac{p(x|t')}{p(x|t)} dx \\
&= \int p(x|t')[\log p(x|t') - \log p(x|t)] dx \\
&= \int p(x|t')[\langle t', f(x)\rangle - \log Z(t') - (\langle t, f(x)\rangle - \log Z(t))] dx \\
&= \int p(x|t')\langle t', f(x)\rangle dx - \log Z(t') - \int p(x|t')\langle t, f(x)\rangle dx + \log Z(t) \\
&= \langle t', \nabla_{t'} \log Z(t')\rangle - \log Z(t') - \langle t, \nabla_{t'} \log Z(t')\rangle + \log Z(t) \\
&= \langle t' - t, \nabla_{t'} \log Z(t')\rangle + \log Z(t) - \log Z(t') \\
&= D_B(t,t')
\end{aligned}$$

where we use the property that the expectation of $f(x)$ under the distribution $p(x|t')$ is equal to the gradient of the log-partition function $\mathbb{E}_{x \sim p(x|t')}[f(x)] = \nabla_{t'} \log Z(t')$.

Thus, we have shown that the limit of the posterior is related to the Bregman divergence of $t'$ and $t$, and is consistent with the theorem statement, which concludes the proof.

$\square$

**Lemma A.2.** *Let $X_i : \Omega \to \mathbb{R}^n$ be measurable functions, $s \in S$, $\omega \in \Omega$, $N \in \mathbb{N}$, and assume $\mathbb{E}[X_i] = \mu_x < \infty$ (first moment exists). Define:*

$$\phi(s, N, \omega) = \psi\left(s, \frac{1}{N}\sum_{i=1}^{N} X_i(\omega)\right), \tag{48}$$

*where $\psi(x, s)$ is a continuous function in both variables and uniformly continuous in $x$ for all $s$. Then:*

$$I \overset{def}{=} \lim_{N \to \infty} \frac{1}{N} \log\left(\int_S e^{N\psi(s, \frac{1}{N}\sum_{i=1}^{N} X_i)} ds\right) \overset{a.s.}{=} \max_{s \in S} \psi(s, \mu_x) \overset{def}{=} M. \tag{49}$$

*Proof of Lemma.* Let $\bar{X}_N = \frac{1}{N}\sum_{i=1}^{N} X_i(\omega)$. By Kolmogorov's strong law of large numbers, we have:

$$\bar{X}_N(\omega) \overset{a.s.}{\longrightarrow} \mu_x, \quad p\left(\{\omega \in \Omega : \lim_{N \to \infty} \bar{X}_N(\omega) = \mu_x\}\right) = 1. \tag{50}$$

Since $\psi(s, x)$ is continuous in $x$, by the continuous mapping theorem:

$$\lim_{N \to \infty} \bar{X}_N(\omega) \overset{a.s.}{\longrightarrow} \mu_x \implies \lim_{N \to \infty} \psi(s, \bar{X}_N(\omega)) \overset{a.s.}{\longrightarrow} \psi(s, \mu_x). \tag{51}$$

Thus, for all $\omega \in \tilde{\Omega}$:

$$\lim_{N \to \infty} \psi(s, \bar{X}_N(\omega)) = \psi(s, \mu_x). \tag{52}$$

In other words, for any $\epsilon > 0$ and fixed $s \in S$, there exists $N_0(\epsilon, s)$ such that for all $N > N_0(\epsilon, s)$:

$$|\psi(s, \bar{X}_N(\omega)) - \psi(s, \mu_x)| < \epsilon, \tag{53}$$

and for all $\omega \in \tilde{\Omega}$, $s \in S$, and $\epsilon > 0$, there exists $N_0(\omega, s, \epsilon)$ such that for all $N > N_0(\omega, s, \epsilon)$:

$$|\psi(s, \bar{X}_N(\omega)) - \psi(s, \mu_x)| < \epsilon. \tag{54}$$

Define:

$$M_N(\omega) = \max_{s \in S} \psi(s, \bar{X}_N(\omega)). \tag{55}$$

Since $\psi(s, x)$ is continuous in $s$ and $S$ is a compact domain, $M_N(\omega) < \infty$. Moreover, since $\bar{X}_N(\omega) \overset{a.s.}{\longrightarrow} \mu_x$ and both $\psi$ and $\max$ are continuous functions:

$$M_N(\omega) = \max_{s \in S} \psi(s, \bar{X}_N(\omega)) \overset{a.s.}{\longrightarrow} \max_{s \in S} \psi(s, \mu_x) = M. \tag{56}$$

Now consider:

$$\int_S e^{N\psi(s, \bar{X}_N(\omega))} ds \le \int_S e^{N \max_{s \in S} \psi(s, \bar{X}_N(\omega))} ds = e^{N \max_{s \in S} \psi(s, \bar{X}_N(\omega))} \text{Vol}(S). \tag{57}$$

Taking the logarithm and dividing by $N$:

$$\frac{1}{N} \log\left(e^{N M_N(\omega)} \text{Vol}(S)\right) \le M_N(\omega) + \frac{1}{N} \log \text{Vol}(S). \tag{58}$$

Finally, taking the limit as $N \to \infty$:

$$\lim_{N \to \infty} \frac{1}{N} \log \int_S e^{N\psi(s, \bar{X}_N(\omega))} ds \leq \lim_{N \to \infty} \left( M_N(\omega) + \frac{1}{N} \log \text{Vol}(S) \right) = M_N(\omega) \xrightarrow{\text{a.s.}} M. \tag{59}$$

Thus, the result holds almost surely:

$$\lim_{N \to \infty} \frac{1}{N} \log \int_S e^{N\psi(s, \bar{X}_N(\omega))} ds \leq M = \max_{s \in S} \psi(s, \mu_x). \tag{60}$$

Now, almost everywhere, we have $I \leq M$. Define:

$$K_\epsilon = \{s \in S : |\psi(s, \mu_x) - M| < \epsilon/2\}. \tag{61}$$

Then:

$$\int_S e^{N\psi(s, \bar{X}_N(\omega))} ds \geq \int_{K_\epsilon} e^{N\psi(s, \bar{X}_N(\omega))} ds \quad \forall N, \forall \omega. \tag{62}$$

Now, consider:

$$|\psi(s, \bar{X}_N(\omega)) - M| = |\psi(s, \bar{X}_N(\omega)) - \psi(s, \mu_x) + \psi(s, \mu_x) - M| \tag{63}$$

$$\leq |\psi(s, \bar{X}_N(\omega)) - \psi(s, \mu_x)| + |\psi(s, \mu_x) - M|. \tag{64}$$

**First Term**: By the strong law of large numbers and uniform continuity of $\psi$, for any $\epsilon > 0$, there exists $N_0$ such that for all $N > N_0$ and $s \in K_\epsilon$:

$$|\psi(s, \bar{X}_N(\omega)) - \psi(s, \mu_x)| < \epsilon/2. \tag{65}$$

This follows because $\bar{X}_N(\omega) \xrightarrow{\text{a.s.}} \mu_x$, and $\psi(s, x)$ is continuous in $x$. Hence, for sufficiently large $N$, $\psi(s, \bar{X}_N(\omega))$ converges uniformly to $\psi(s, \mu_x)$ for $s \in K_\epsilon$.

**Second Term**: By the definition of $K_\epsilon$, for all $s \in K_\epsilon$:

$$|\psi(s, \mu_x) - M| < \epsilon/2. \tag{66}$$

Combining these results, for all $\epsilon > 0$, $s \in K_\epsilon$, and $N > N_0$:

$$|\psi(s, \bar{X}_N(\omega)) - M| < \epsilon. \tag{67}$$

Thus, for all $\epsilon > 0$, $s \in K_\epsilon$, and almost all $\omega \in \Omega$:

$$\psi(s, \bar{X}_N(\omega)) > M - \epsilon. \tag{68}$$

Therefore:

$$\int_{K_\epsilon} e^{N\psi(s, \bar{X}_N(\omega))} ds \geq \int_{K_\epsilon} e^{N(M-\epsilon)} ds = e^{N(M-\epsilon)} \text{Vol}(K_\epsilon). \tag{69}$$

Taking the logarithm and dividing by $N$:

$$\frac{1}{N} \log \int_{K_\epsilon} e^{N\psi(s, \bar{X}_N(\omega))} ds \geq \frac{1}{N} \log \left( e^{N(M-\epsilon)} \text{Vol}(K_\epsilon) \right) \tag{70}$$

$$= M - \epsilon + \frac{1}{N} \log \text{Vol}(K_\epsilon). \tag{71}$$

Taking the limit as $N \to \infty$:

$$\lim_{N \to \infty} \frac{1}{N} \log \int_{K_\epsilon} e^{N\psi(s, \bar{X}_N(\omega))} ds \geq M - \epsilon. \tag{72}$$

Combining this with the upper bound $I \leq M$, we have:

$$M - \epsilon \leq I \leq M. \tag{73}$$

Since this holds for all $\epsilon > 0$, it follows that:

$$I = M. \tag{74}$$

$\square$

**Lemma A.3.** *We will show that the following function is bounded*

$$\phi(s, \omega, N) = \left\langle s, \bar{X}_N(\omega)) \right\rangle - \log Z(s), \tag{75}$$

*where $N \in \mathbf{N}$, $s \in S \subset \mathbb{R}^n$*

*Proof.* First, we use the Cauchy–Schwarz inequality

$$\left| \left\langle s, \bar{X}_N(\omega)) \right\rangle \right| \leq \|s\|_2 \cdot \left\| \bar{X}_N(\omega)) \right\|_2 . \tag{76}$$

Furthermore, since the $L_2$ norm is a continuous function, and there is almost sure convergence $\bar{X}_N \to \mu_x$, then

$$\left\| \bar{X}_N(\omega)) \right\|_2 \xrightarrow{\text{a.s.}} \|\mu_x\|_2. \tag{77}$$

Thus, there exists $N_0$ such that for all $N > N_0$,

$$\left| \left\| \bar{X}_N(\omega)) \right\|_2 - \|\mu_x\|_2 \right| < \epsilon \implies \left\| \bar{X}_N(\omega)) \right\|_2 < \|\mu_x\|_2 + \epsilon. \tag{78}$$

Since $\log Z$ is continuous on the compact set $S$, there exists $C_Z$ such that $\log Z(s) < C_Z$ for all $s \in S$.

On the compact set $S$, there exists $C_s$ such that $\|s\|_2 < C_s$ for all $s \in S$.

Therefore, for all $s \in S$, there exists $N_0$ such that for all $N > N_0$,

$$\phi(s, \omega, N) < C = \|\mu_x\|_2 \cdot C_s + C_Z + \epsilon \tag{79}$$

$\square$

**Lemma A.4.** *We will show that the following function is uniformly continuous*

$$\phi(s, \omega, N) = \left\langle s, \bar{X}_N(\omega)) \right\rangle - \log Z(s), \tag{80}$$

*where $N \in \mathbf{N}$, $s \in S \subset \mathbb{R}^n$*

*Proof.* Let $x_1$ and $x_2$ be vectors in $\mathbb{R}^n$.

$$\begin{aligned}
|\psi(s, x_1) - \psi(s, x_2)| &= |\langle s, x_1 \rangle - \log Z(s) - (\langle s, x_2 \rangle - \log Z(s))| \\
&= |\langle s, x_1 \rangle - \langle s, x_2 \rangle| \\
&= |\langle s, x_1 - x_2 \rangle| \\
&= |s| \, |x_1 - x_2| \, |\cos(\theta)|
\end{aligned}$$

where $\theta$ is the angle between $s$ and $x_1 - x_2$. Since $S$ is compact, there exists $M$ such that $|s| < M$ for all $s \in S$. Thus,

$$|\psi(s, x_1) - \psi(s, x_2)| \leq M |x_1 - x_2|$$

Given $\varepsilon > 0$, we choose $\delta = \varepsilon / M$. If $|x_1 - x_2| < \delta$, then

$$|\psi(s, x_1) - \psi(s, x_2)| < \varepsilon$$

Therefore, $\psi(s, x)$ is uniformly continuous in $x$. $\square$

**Theorem A.5.** *Suppose that the following integral converges to zero*

$$\int_\Theta \int_\Theta \left| e^{-D_{\log Z_1(t)}(t',t)} - e^{-D_{\log Z_2(t)}(t',t)} \right|^2 dt\, dt' \to 0, \tag{81}$$

*where*

$$D_{\varphi(t)}(t',t) = \varphi(t') - \varphi(t) - \langle \nabla_{t'}\varphi(t'), t' - t\rangle$$

*is the Bregman divergence. Then there $\log Z_1(t)$ converges to $\log Z_2(t)$ uniformly in $t$*

$$\|\nabla^2 \log Z_1(t) - \nabla^2 \log Z_2(t)\| \to 0, \tag{82}$$

*where $\|\cdot\|$ denotes the $L^2$ norm.*

*Proof.* Since the integrand is non-negative and its integral goes to zero, the integrand must converge to zero almost everywhere. As the functions involved are continuous, we have:

$$e^{-D_{\log Z_1(t)}(t',t)} = e^{-D_{\log Z_2(t)}(t',t)} \tag{83}$$

for all $t, t' \in \Theta$. Taking the logarithm of both sides, we get:

$$-D_{\log Z_1(t)}(t',t) = -D_{\log Z_2(t)}(t',t) \tag{84}$$

which implies

$$D_{\log Z_1(t)}(t',t) = D_{\log Z_2(t)}(t',t). \tag{85}$$

Substituting the definition of the Bregman divergence, we have:

$$\log Z_1(t') - \log Z_1(t) - \langle \nabla \log Z_1(t), t' - t\rangle$$
$$= \log Z_2(t') - \log Z_2(t) - \langle \nabla \log Z_2(t), t' - t\rangle.$$

Rearranging terms gives:

$$(\log Z_1(t') - \log Z_2(t')) - (\log Z_1(t) - \log Z_2(t)) - \langle \nabla \log Z_1(t) - \nabla \log Z_2(t), t' - t\rangle = 0. \tag{86}$$

Let $\Delta(t) = \log Z_1(t) - \log Z_2(t)$. Then the equation becomes:

$$\Delta(t') - \Delta(t) - \langle \nabla\Delta(t), t' - t\rangle = 0. \tag{87}$$

This implies that the first-order Taylor expansion of $\Delta(t')$ around $t$ is exact, which holds if and only if $\Delta(t)$ is an affine function:

$$\Delta(t) = \langle c, t\rangle + b, \tag{88}$$

for some constant vector $c$ and scalar $b$. It means that two log-partition function are related via an affine transformation

$$\log Z_1(t) = \log Z_2 - \langle c, t\rangle + b \tag{89}$$

Therefore the $L^2$ norm equals to zero

$$\|\nabla^2 \log Z_1(t) - \nabla^2 \log Z_2(t)\| = 0, \tag{90}$$

$$\square$$

**Lemma A.6** (Vanishing Gradient in Bregman MSE Loss). *Consider the loss functional $L = \int_S \int_S \left| e^{-D_{\log Z_1(t)}(t,t')} - e^{-D_{\log Z_2(t)}(t,t')} \right|^2 dt\, dt'$. Assume there exists a region $R \subset S \times S$ where either $D_{\log Z_1(t)}(t,t') \geq C$ or $D_{\log Z_2(t)}(t,t') \geq C$ uniformly for some $C \gg 1$. Then the gradient of $L$ with respect to parameters governing $\log Z_1(t)$ satisfies:*

$$\|\nabla L\| \leq Ke^{-C}, \tag{91}$$

*where $K > 0$ is a constant which depends on the measure of $R$ and Lipschitz constant of $\log Z_1(t)$ which we assume to be bounded. Thus, gradients vanish exponentially as $C$ increases.*

*Proof.* The gradient of the loss functional $L$ with respect to $\log Z_1(t)$ can be formally written (using functional derivatives) as proportional to:

$$\nabla L \propto \int_S \int_S \frac{\delta}{\delta \log Z_1(t)} \left| e^{-D_{\log Z_1(t)}(t,t')} - e^{-D_{\log Z_2(t)}(t,t')} \right|^2 dt \, dt'. \tag{92}$$

Expanding the square and taking the derivative with respect to $\log Z_1(t)$, we focus on the terms involving $D_{\log Z_1(t)}(t,t')$. The gradient is then proportional to:

$$\nabla L \propto \int_S \int_S \left( e^{-D_{\log Z_1(t)}(t,t')} - e^{-D_{\log Z_2(t)}(t,t')} \right) \frac{\delta}{\delta \log Z_1(t)} e^{-D_{\log Z_1(t)}(t,t')} dt \, dt'. \tag{93}$$

The functional derivative of $e^{-D_{\log Z_1(t)}(t,t')}$ with respect to $\log Z_1(t)$ involves the derivative of the Bregman divergence $D_{\log Z_1(t)}(t,t')$ with respect to $\log Z_1(t)$. Let us denote $\mathcal{D}_1(t,t') = D_{\log Z_1(t)}(t,t')$ and $\mathcal{D}_2(t,t') = D_{\log Z_2(t)}(t,t')$. Then the gradient can be expressed as:

$$\nabla L \propto \int_S \int_S \left( e^{-\mathcal{D}_1(t,t')} - e^{-\mathcal{D}_2(t,t')} \right) e^{-\mathcal{D}_1(t,t')} (-\nabla \mathcal{D}_1(t,t')) dt \, dt', \tag{94}$$

where $\nabla \mathcal{D}_1(t,t')$ represents the gradient of $D_{\log Z_1(t)}(t,t')$ with respect to parameters governing $\log Z_1(t)$. We consider the magnitude of the integrand within the region $R \subset S \times S$. In region $R$, either $D_{\log Z_1(t)}(t,t') \geq C$ or $D_{\log Z_2(t)}(t,t') \geq C$.

Case 1: $D_{\log Z_1(t)}(t,t') \geq C$. In this case, $e^{-D_{\log Z_1(t)}(t,t')} \leq e^{-C}$. The magnitude of the integrand is bounded by:

$$\left| \left( e^{-D_{\log Z_1(t)}(t,t')} - e^{-D_{\log Z_2(t)}(t,t')} \right) e^{-D_{\log Z_1(t)}(t,t')} (-\nabla \mathcal{D}_1(t,t')) \right|$$

$$\leq \left( |e^{-D_{\log Z_1(t)}(t,t')}| + |e^{-D_{\log Z_2(t)}(t,t')}| \right) |e^{-D_{\log Z_1(t)}(t,t')}| \|\nabla \mathcal{D}_1(t,t')\|$$

$$\leq \left( e^{-C} + e^{-D_{\log Z_2(t)}(t,t')} \right) e^{-C} \|\nabla \mathcal{D}_1(t,t')\|$$

$$\leq (1 + e^{-D_{\log Z_2(t)}(t,t')}) e^{-C} \|\nabla \mathcal{D}_1(t,t')\|.$$

Assuming that $\|\nabla \mathcal{D}_1(t,t')\|$ is bounded by some constant $M'$ due to finite Lipschitz constant of $\log Z_1(t)$ and since $e^{-D_{\log Z_2(t)}(t,t')} \leq 1$ (for non-negative Bregman divergences), the integrand magnitude is bounded by $2M'e^{-C}$.

Case 2: $D_{\log Z_2(t)}(t,t') \geq C$. In this case, $e^{-D_{\log Z_2(t)}(t,t')} \leq e^{-C}$. The magnitude of the integrand is bounded by:

$$\left| \left( e^{-D_{\log Z_1(t)}(t,t')} - e^{-D_{\log Z_2(t)}(t,t')} \right) e^{-D_{\log Z_1(t)}(t,t')} (-\nabla \mathcal{D}_1(t,t')) \right|$$

$$\leq \left( |e^{-D_{\log Z_1(t)}(t,t')}| + |e^{-D_{\log Z_2(t)}(t,t')}| \right) |e^{-D_{\log Z_1(t)}(t,t')}| \|\nabla \mathcal{D}_1(t,t')\|$$

$$\leq \left( e^{-D_{\log Z_1(t)}(t,t')} + e^{-C} \right) e^{-D_{\log Z_1(t)}(t,t')} \|\nabla \mathcal{D}_1(t,t')\|$$

$$= \left( e^{-2D_{\log Z_1(t)}(t,t')} + e^{-C} e^{-D_{\log Z_1(t)}(t,t')} \right) \|\nabla \mathcal{D}_1(t,t')\|$$

$$\leq (1+1) e^{-C} M' = 2M'e^{-C},$$

assuming $D_{\log Z_1(t)}(t,t') \geq 0$ and $e^{-D_{\log Z_1(t)}(t,t')} \leq 1$.

In both cases, within the region $R$, the integrand's magnitude is bounded by $2M'e^{-C}$. Let $m(R)$ be the measure of region $R$. Then the norm of the gradient can be bounded by integrating over $R$:

$$\|\nabla L\| \leq \int_R 2M'e^{-C} dt \, dt' = 2M'e^{-C} m(R) = Ke^{-C}, \tag{95}$$

where $K = 2M'm(R)$ is a constant dependent on the measure of $R$ and the bound $M'$ which is related to the Lipschitz properties of $\log Z_1(t)$. This shows that the gradient vanishes exponentially as $C$ increases. $\qquad \square$

**Lemma A.7.** *Let* $Z : S \to (0,\infty)$ *with* $S \subset \mathbb{R}^n$ *compact domain and* $\log Z$ *convex. For any fixed* $t \in S$:

$$\frac{\exp\left( -D_{\log Z}(t',t) \right)}{\int_S \exp\left( -D_{\log Z}(s,t) \right) ds} = \frac{\exp\left( -\langle t', \nabla \log Z(t) \rangle + \log Z(t') \right)}{\int_S \exp\left( -\langle s, \nabla \log Z(t) \rangle + \log Z(s) \right) ds} \tag{96}$$

*Proof.* Expand the Bergman divergence $D_{\log Z}(t', t) = \log Z(t') - \log Z(t) - \langle \nabla \log Z(t), t' - t \rangle$. The numerator can be rewritten as:

$$\begin{aligned}
\exp(-D_{\log Z}(t', t)) &= \exp\left(-\left(\log Z(t') - \log Z(t) - \langle \nabla \log Z(t), t' - t \rangle\right)\right) \\
&= \exp\left(-\log Z(t') + \log Z(t) + \langle \nabla \log Z(t), t' \rangle - \langle \nabla \log Z(t), t \rangle\right) \\
&= \exp(\log Z(t)) \exp(-\langle \nabla \log Z(t), t \rangle) \exp(-\log Z(t') + \langle \nabla \log Z(t), t' \rangle)
\end{aligned} \tag{97}$$

The denominator, integrating over $s \in S$, becomes:

$$\begin{aligned}
\int_S \exp(-D_{\log Z}(s, t)) \, ds &= \int_S \exp\left(-\left(\log Z(s) - \log Z(t) - \langle \nabla \log Z(t), s - t \rangle\right)\right) \, ds \\
&= \int_S \exp\left(-\log Z(s) + \log Z(t) + \langle \nabla \log Z(t), s \rangle - \langle \nabla \log Z(t), t \rangle\right) \, ds \\
&= \exp(\log Z(t)) \exp(-\langle \nabla \log Z(t), t \rangle) \int_S \exp(-\log Z(s) + \langle \nabla \log Z(t), s \rangle) \, ds \\
&= Z(t) \exp(-\langle \nabla \log Z(t), t \rangle) \int_S \frac{\exp(\langle \nabla \log Z(t), s \rangle)}{Z(s)} \, ds
\end{aligned} \tag{98}$$

Forming the ratio of the numerator (97) and the denominator (98):

$$\frac{\exp(-D_{\log Z}(t', t))}{\int_S \exp(-D_{\log Z}(s, t)) \, ds} = \frac{Z(t) \exp(-\langle \nabla \log Z(t), t \rangle) \frac{\exp(\langle \nabla \log Z(t), t' \rangle)}{Z(t')}}{Z(t) \exp(-\langle \nabla \log Z(t), t \rangle) \int_S \frac{\exp(\langle \nabla \log Z(t), s \rangle)}{Z(s)} \, ds} \tag{99}$$

Simplifying the ratio:

$$\frac{\exp(-D_{\log Z}(t', t))}{\int_S \exp(-D_{\log Z}(s, t)) \, ds} = \frac{\frac{\exp(\langle \nabla \log Z(t), t' \rangle)}{Z(t')}}{\int_S \frac{\exp(\langle \nabla \log Z(t), s \rangle)}{Z(s)} \, ds} \tag{100}$$

Rewrite the terms in the exponent:

$$\frac{\exp(\langle \nabla \log Z(t), x \rangle)}{Z(x)} = \exp(\langle \nabla \log Z(t), x \rangle - \log Z(x)) = \exp(-\log Z(x) + \langle \nabla \log Z(t), x \rangle) \tag{101}$$

Multiplying the exponent by -1 and rearranging:

$$\frac{\exp(\langle \nabla \log Z(t), x \rangle)}{Z(x)} = \exp(-(\log Z(x) - \langle \nabla \log Z(t), x \rangle)) = \exp(-(-\langle x, \nabla \log Z(t) \rangle + \log Z(x))) \tag{102}$$

Further manipulation:

$$\frac{\exp(\langle \nabla \log Z(t), x \rangle)}{Z(x)} = \exp(-\langle x, \nabla \log Z(t) \rangle + \log Z(x)) \tag{103}$$

Substituting this back into equation (100):

$$\frac{\exp(-D_{\log Z}(t', t))}{\int_S \exp(-D_{\log Z}(s, t)) \, ds} = \frac{\exp(-\langle t', \nabla \log Z(t) \rangle + \log Z(t'))}{\int_S \exp(-\langle s, \nabla \log Z(t) \rangle + \log Z(s)) \, ds} \tag{104}$$

This yields exactly the RHS expression. $\qquad\square$

**Proposition A.8.** *Suppose that the (target) data distribution is a bimodal mixture of two Gaussians, each with variance $\sigma^2$:*

$$p_0(x) = \frac{1}{2}\mathcal{N}(x \mid -1, \sigma^2) + \frac{1}{2}\mathcal{N}(x \mid 1, \sigma^2). \tag{105}$$

*The latent (source) distribution is the standard normal $\mathcal{N}(x \mid 0, 1)$. Consider the variance-preserving SDE*

$$dX_t = -\frac{1}{2}\beta X_t \, dt + \sqrt{\beta} \, dW_t. \tag{106}$$

*Then the Lyapunov exponent of the corresponding reverse-time ODE at $x = 0$ has the following form:*

$$\lambda = \frac{\beta}{2}\left(1 + \frac{1 - \sigma^2}{\sigma^4}\right), \tag{107}$$

*and it diverges to infinity as $\sigma \to 0$. In this case, the point $x = 0$ can be interpreted as a phase transition boundary.*

*Proof.* We begin with the reverse probability flow ODE:

$$\frac{dX_s}{ds} = -f(X_s, t) + \frac{g(t)^2}{2}\nabla_x \log p_t(X_s), \tag{108}$$

where the drift term is:

$$v(x) = -f(x, t) + \frac{g(t)^2}{2}\nabla_x \log p_t(x). \tag{109}$$

Linearizing around $x = 0$, the Lyapunov exponent is defined as:

$$\lambda = v'(0) = -f'(0, t) + \frac{g(t)^2}{2}\frac{d^2}{dx^2}\log p_t(x). \tag{110}$$

The density $p_t(x)$ is computed via convolution with the Gaussian noising kernel:

$$p_t(x) = \int_{-\infty}^{\infty} p_0(y)\,\mathcal{N}\left(x \mid e^{-\frac{1}{2}\beta t}y,\ 1 - e^{-\beta t}\right)dy. \tag{111}$$

Since convolution of Gaussians is still Gaussian, we obtain:

$$p_t(x) = \frac{1}{2\sqrt{2\pi}\sigma_1(t)}A(x), \tag{112}$$

where

$$A(x) = \exp\left(-\frac{(x - \mu(t))^2}{2\sigma_1^2(t)}\right) + \exp\left(-\frac{(x + \mu(t))^2}{2\sigma_1^2(t)}\right), \tag{113}$$

and

$$\sigma_1^2(t) = e^{-\beta t}\sigma^2 + (1 - e^{-\beta t}), \quad \mu(t) = e^{-\frac{1}{2}\beta t}. \tag{114}$$

Computing the derivative in the definition of the Lyapunov exponent, we get:

$$\lambda = \frac{\beta}{2} + \frac{\beta}{2}\cdot\frac{e^{-\beta t} - \sigma_1^2(t)}{\sigma_1^4(t)} = \frac{\beta}{2}\left(1 + \frac{e^{-\beta t} - \sigma_1^2(t)}{\sigma_1^4(t)}\right). \tag{115}$$

As time goes to 0, we have $\sigma_1 \to \sigma$, and thus:

$$\lambda = \frac{\beta}{2}\left(1 + \frac{1 - \sigma^2}{\sigma^4}\right), \tag{116}$$

suggesting divergence of nearby reverse ODE trajectories for small $\sigma$, and identifying $x = 0$ as a potential phase transition boundary. $\qquad\square$

## B. Experimental Details

### B.1. Numerics

**Mean-as-Stat** The main idea of posterior-mean-as- statistics is to predict the parameters $t$ based on the microstate $s$ by minimizing the regression error:

$$L(\phi) = \mathbb{E}_{s\sim\mathbb{P}(s|t),t\sim\mathbb{P}(t)}\|f_\phi(s) - t\|_2^2. \tag{117}$$

A function $f_\phi$ learned in this way will predict the mean of the posterior distribution $\mathbb{P}(t|s)$.

To evaluate the quality of the free energy reconstruction, we compute the RMSE between the ground truth free energy $F_{gt}(t)$ and the reconstructed free energy $F_{rec}(t)$:

$$\mathrm{RMSE}(F_{rec}, F_{gt}) = \min_A \|F_{gt}(t) - A(F_{rec}(t))\|_2^2, \tag{118}$$

where we minimize over all possible affine transformations $A$, accounting for the fact that free energy is only defined up to such a transformation

**Ising Model.** Since the Ising model lacks an exact free energy solution for $H \neq 0$, we construct the ground truth $F_{\mathrm{ising}}(T, H)$ by numerically integrating the known magnetization $M(T, H)$ and energy $E(T, H)$. The ground truth free energy $F_{ising}(T, H)$ must satisfy the following partial derivative relations with respect to temperature and magnetic field::

$$\frac{\partial F_{\mathrm{ising}}(T, H)}{\partial T} = E(T, H), \quad \frac{\partial F_{\mathrm{ising}}(T, H)}{\partial H} = M(T, H). \tag{119}$$

To approximate $F_{\mathrm{ising}}(T, H)$, we train a feedforward neural network $F_\theta(T, H)$ on the domain $H > 0$, where the free energy is $C^2$-smooth. The approximation is extended to $H < 0$ using the symmetry $F_{\mathrm{ising}}(T, -H) = F_{\mathrm{ising}}(T, H)$. The loss function enforces consistency with the partial derivatives:

$$\mathcal{L}(\theta) = \int_\Omega \left( \left\| \frac{\partial F_\theta}{\partial T} - E(T, H) \right\|_2^2 + \left\| \frac{\partial F_\theta}{\partial H} - M(T, H) \right\|_2^2 \right) dT dH. \tag{120}$$

For the posterior-mean-as-sufficient-statistics method, we adopt a similar approach to integrate the predicted sufficient statistics $s_T(T, H)$ and $s_H(T, H)$ by minimizing the loss:

$$\mathcal{L}(\theta) = \int_\Omega \left( \left\| \frac{\partial F_\theta}{\partial T} - s_T(T, H) \right\|_2^2 + \left\| \frac{\partial F_\theta}{\partial H} - s_H(T, H) \right\|_2^2 \right) dT dH. \tag{121}$$

After obtaining the ground truth free energy, along with the free energy estimated by our Bayesian thermodynamic integration and the posterior-mean-as-statistics method, we minimize the RMSE with respect to the ground truth over affine transformations to account for the fact that free energy is only defined up to an affine transformation. Finally, we evaluate and compare the resulting errors with the baseline.

**TASEP Model.** For TASEP, which has an analytical free energy solution, we use a similar neural network approach. The ground truth free energy is given by:

$$F_{\mathrm{TASEP}}(\alpha, \beta) = \begin{cases} \frac{1}{4}, & \alpha > \frac{1}{2}, \ \beta > \frac{1}{2}; \\ \alpha(1 - \alpha), & \alpha < \beta, \ \alpha < \frac{1}{2}; \\ \beta(1 - \beta), & \beta < \alpha, \ \beta < \frac{1}{2}. \end{cases} \tag{122}$$

The network $F_\theta(\alpha, \beta)$ is trained to minimize:

$$\mathcal{L}(\theta) = \int_\Omega \left( \left\| \frac{\partial F_\theta}{\partial \alpha} - J_\alpha(\alpha, \beta) \right\|_2^2 + \left\| \frac{\partial F_\theta}{\partial \beta} - J_\beta(\alpha, \beta) \right\|_2^2 \right) d\alpha d\beta, \tag{123}$$

where $J_\alpha$ and $J_\beta$ are the particle currents at the boundaries, analogous to energy/magnetization in the Ising model. Finally, the resulting free energy surface is evaluated using the RMSE between the true TASEP free energy and the reconstructed free energy, after accounting for possible affine transformations:

$$\mathrm{RMSE}(F_{rec}, F_{gt}) = \min_A |F_{gt}(\alpha, \beta) - A(F_{rec}(\alpha, \beta))|_2^2. \tag{124}$$

## B.2. Datasets

**Ising Model.** Our dataset consists of $N = 5.4 \times 10^5$ samples of spin configurations on the square lattice of size $L \times L = 128 \times 128$ with periodic boundary conditions. We consider the parameter ranges $\beta^{-1} = T \in [T_{\min}, T_{\max}] = [1, 5]$, $H \in [H_{\min}, H_{\max}] = [-2, 2]$ similar to the ranges used in (Walker, 2019). Point $(T, H)$ is sampled uniformly from this rectangle, and then a sample spin configuration is created for these values of temperature and external field by starting with a random initial condition and equilibrating is with Glauber (one-spin Metropolis) dynamics (see, e.g. (Krapivsky et al., 2010)) for $10^4 \times 128 \times 128 \approx 1.64 \times 10^8$ iterations. We represent spin configuration as a single-channel image with color of each pixel taking values $+1$ and $-1$. When constructing target probability distributions we choose $\sigma = \frac{1}{50}$ and set the discretization $\mathcal{D}$ of the square $[T_{\min}, T_{\max}] \times [H_{\min}, H_{\max}] = [1, 5] \times [-2, 2]$ to be a uniform grid with $L \times L = 128 \times 128$ grid cells.

Image-to-image network with U$^2$-Net architecture (Qin et al., 2020) is used to approximate posterior $p_\theta(t|s)$. The network takes as input a bundle of $K_{\text{bundle}}$ images concatenated across channel dimension and outputs a categorical distribution representing density values in discrete grid points. For simplicity we choose the discretization $\mathcal{D}$ to be of the same spatial dimensions as the input image. For all our numerical experiments the training was performed on a single Nvidia-HGX compute node with 8 A100 GPUs. We trained U$^2$-Net using Adam optimizer with learning rate 0.00001 and batch size of 2048 for $N_{\text{U2Net steps}} = 20000$ gradient update steps. In all our experiments the training set consists of 80% of samples and the other 20% are used for testing.

**TASEP Model.** We generate a dataset of $N = 150000$ stationary TASEP configurations on a 1d lattice with $M = 16384$ sites. The rates $\alpha(\beta)$ of adding (removing) particles at the left(right) boundary are sampled from the uniform prior distribution over a square $[0, 1] \times [0, 1]$. To reach the stationary state we start from a random initial condition and perform $N_{\text{steps}} = 2 \times 10^9 \approx 8M^2$ move attempts, which is known to be enough to achieve the stationary state except for the narrow vicinity of the transition line $\alpha = \beta < 1/2$ between high-density and low-density phases (in this case the stationary state has a slowly diffusing front of a shock wave in it, one needs of order $M^2$ move attempts to form the shock but of order $M^3$ move attempts for it to diffusively explore all possible positions).

We reshape 1d lattice with $M = 16384$ sites into an image of size $L \times L = 128 \times 128$ using raster scan ordering. To construct target probability distributions we set $\sigma = \frac{1}{150}$ and define the discretization $\mathcal{D}$ as a uniform grid on $[\alpha_{\min}, \alpha_{\max}] \times [\beta_{\min}, \beta_{\max}] = [0, 1] \times [0, 1]$ with $L \times L = 128 \times 128$ grid cells.

## B.3. U$^2$Net Training

Our task now is to estimate the posterior distribution $p(t|s)$ using the set of samples $t_k, s_k$. To do that, we are trying to approximate $p(t|s)$ by representatives from some parametric family of distributions $p_\theta(\mathbf{x}|y)$, choosing $\theta$ to maximize some cost function. It is conventional to choose the cost function to maximize the likelihood of the true external parameters $t_i$ given $s_i$ over all samples in the set:

$$\text{NegativeLogLikelihood}(\theta) = -\sum_{i=1}^{N} \log(p_\theta(t_i|s_i)) \approx$$

$$\approx N\mathbb{E}_{\{t', s\} \sim p(t')p(s|t')} \log(p_\theta(t|s)), \tag{125}$$

where on the right hand side we replaced the summands by their expected values. Minimization of the log-likelihood can be reinterpreted as the minimization of the KL divergence between the target distribution

$$p_{\text{target}}(t|t') = \delta(t - t') \tag{126}$$

(i.e., the reconstructed labels $t$ are identical to the input labels $t'$) and the predicted distribution $\int p_\theta(t|s)p(s|\mathbf{t'})ds$:

$$\mathcal{L}(\theta) = \mathbb{E}_{t' \sim P(t)} \text{KL}\left(p_{\text{target}}(t|t') \middle|\middle| \int p_\theta(t|s)p(s|\mathbf{t'})ds\right). \tag{127}$$

In practice, to avoid divergences it is convenient to replace the "hard label" target distribution (126) with a smoothened distribution

$$p_{\text{target}}(t|t') = C \cdot \exp\left(-\frac{1}{2\sigma^2}||t' - t||^2\right), \tag{128}$$

where $\sigma$ is a smoothening parameter and $C$ is the normalizing constant.

**B.4. Supplementary Figures**

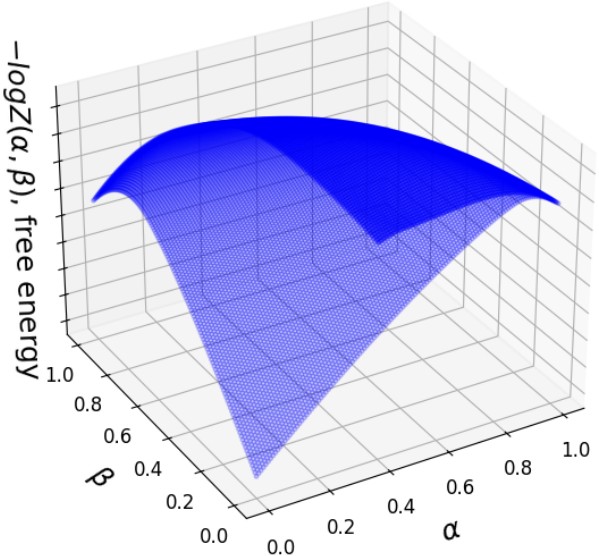
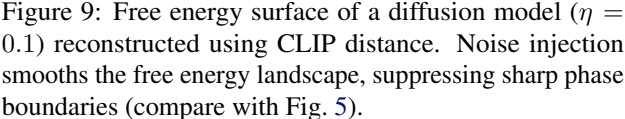

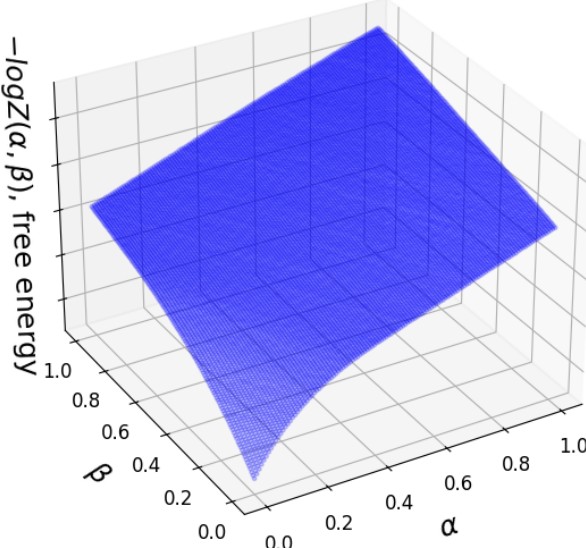

Figure 9: Free energy surface of a diffusion model ($\eta = 0.1$) reconstructed using CLIP distance. Noise injection smooths the free energy landscape, suppressing sharp phase boundaries (compare with Fig. 5).

Figure 10: Free energy surface of a StyleGAN v3 reconstructed using CLIP distance. The convex curvature indicates a single dominant phase, contrasting with the multi-phase structure of diffusion models (left).

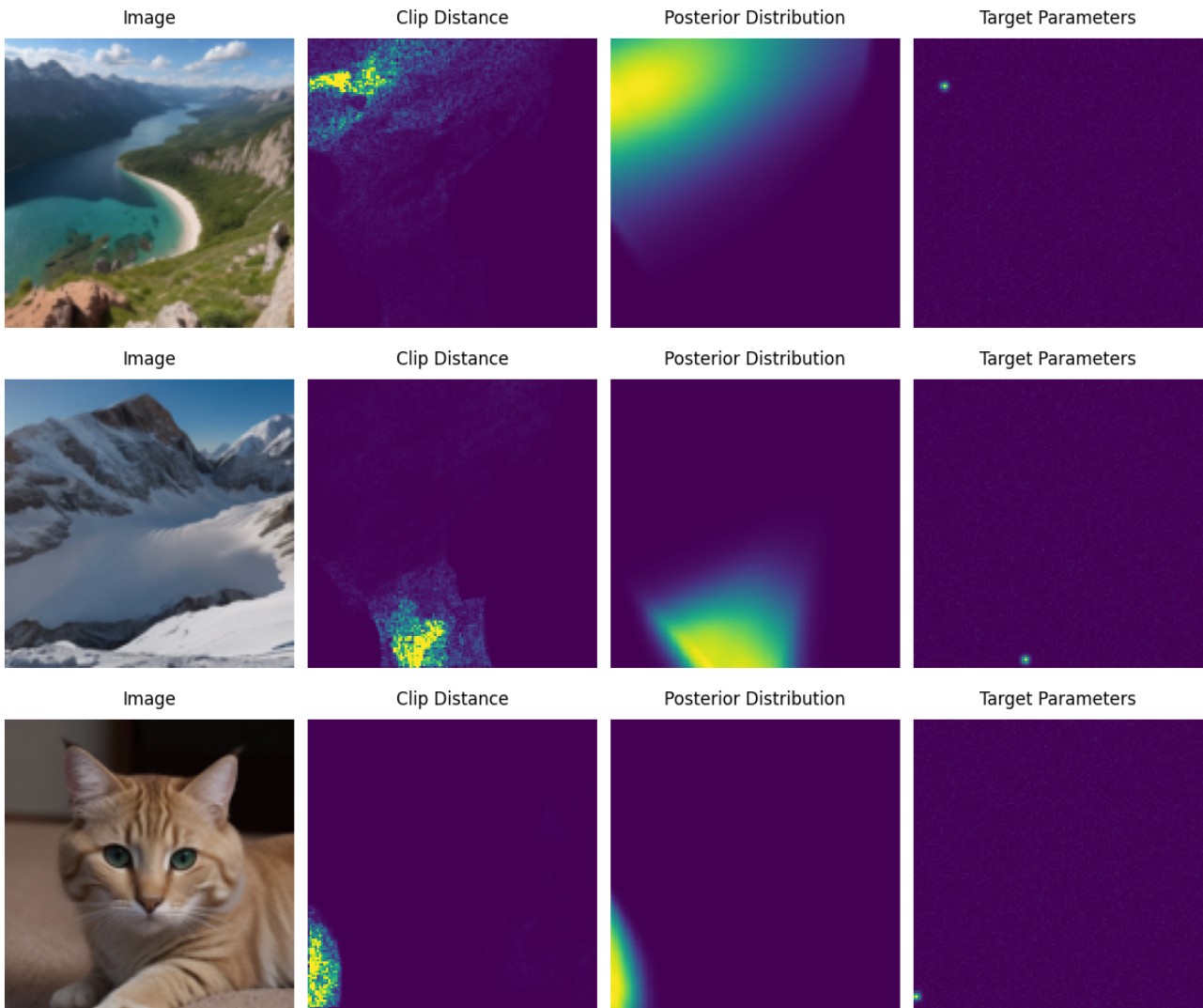

Figure 11: Training examples for free energy reconstruction. Ground truth images with known generation parameters. CLIP-induced latent distribution. Predicted distribution from our convex model.

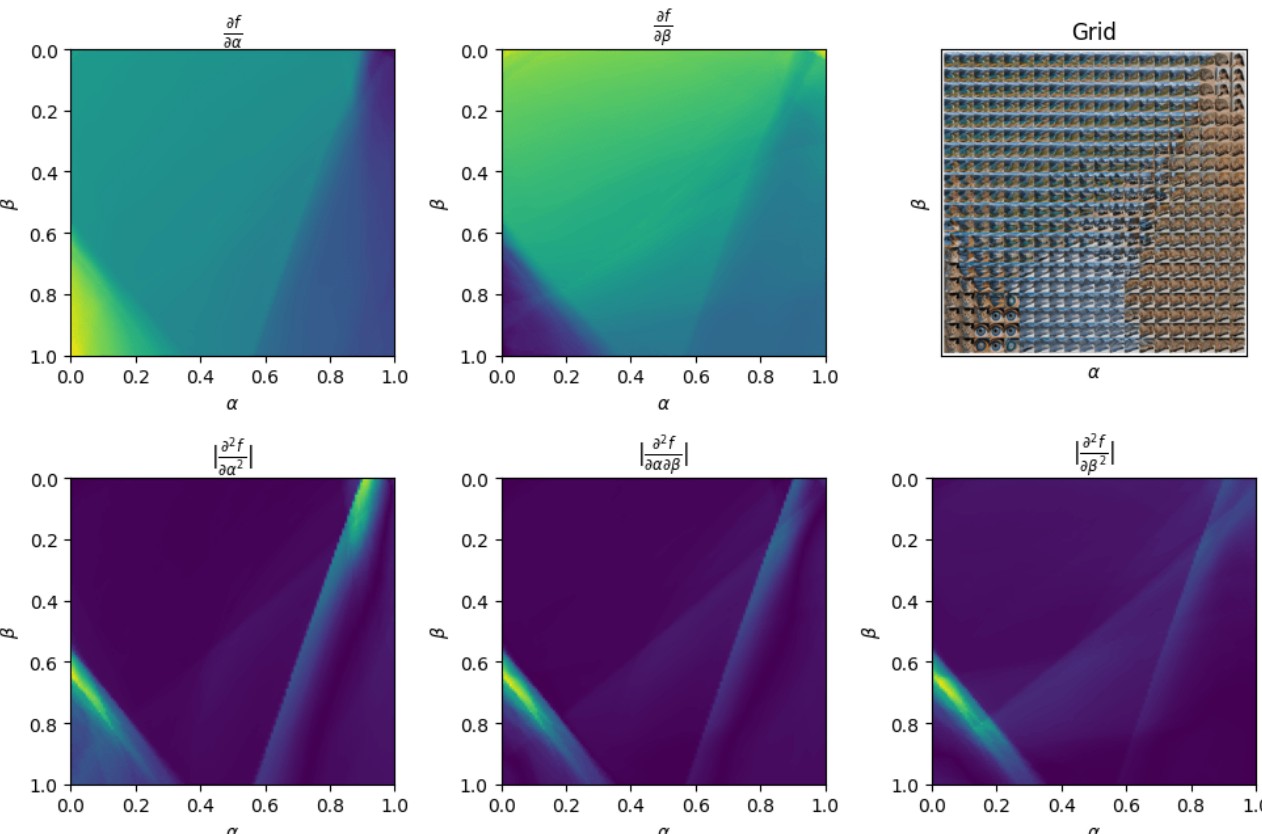

Figure 12: First- and second-order derivatives of the diffusion model's free energy. Discontinuities mark first-order phase transitions and second-order transitions. Regions of constant derivative correlate with visually distinct phases.

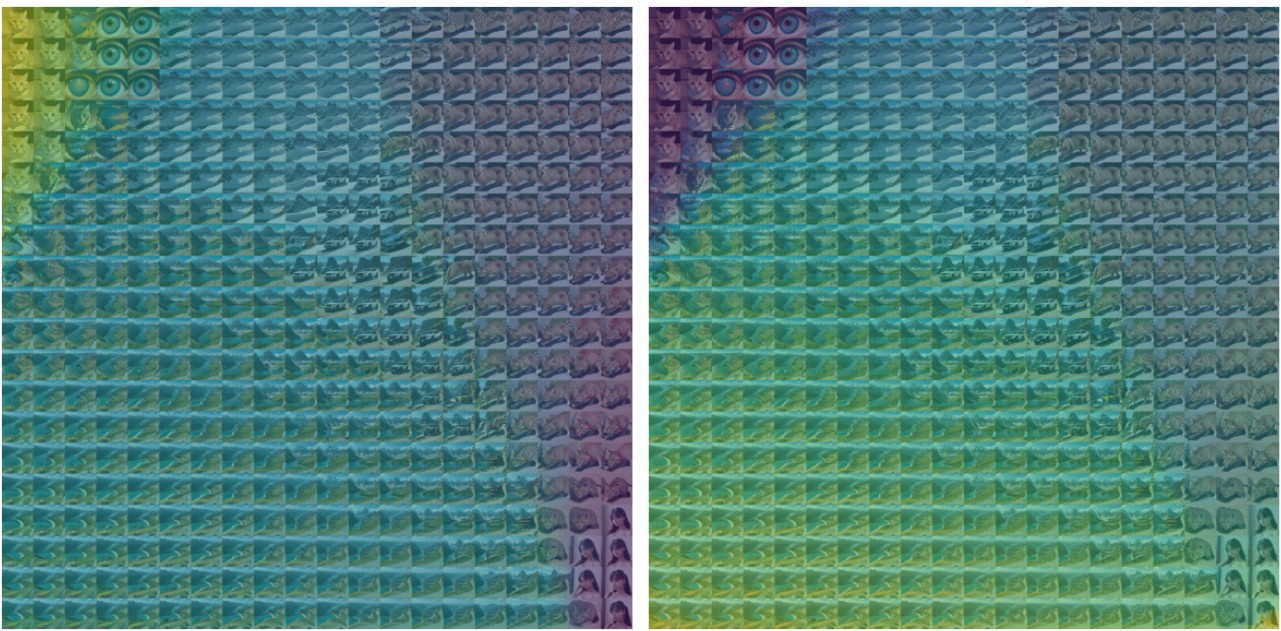

Figure 13: Free energy derivatives of diffusion model's free energy reconstructed with clip distance. Overlaid grid highlights regions of constant derivative, correlating with visually distinct phases.

