# OpenReview forum: "Hessian Geometry of Latent Space in Generative Models"
_ICML.cc/2025/Conference — ICML 2025 poster_

### Official Review · Reviewer_GBhG · 2025-03-09

**Overall Recommendation:** 4

**Summary:**

## Update After Rebuttal
## I maintained my score. Please see my response to the reviewer below for my reasons.
----
This work presents a novel technique for analyzing latent space geometry in diffusion models. Based on the reconstruction of the Fisher Information metric, this method approximates the posterior distribution of latent variables given synthetic samples generated from a diffusion model, and uses this information to learn the log-partition function of the variable $t$. To develop this method, the work relies heavily on the theoretical and mathematical works done in
[Amari and Armstrong (2013)](http://arxiv.org/pdf/1312.1103) and [Bryant (2024)](https://arxiv.org/abs/2405.06998) for its own derivations and assumptions.

When applied to diffusion models, this new method highlights structures of phase transition in the latent space, parameterized by interpolation values $\alpha$ and $\beta$ which represent $t$, allowing the authors to find geodesics or shortest possible paths between latent variables --- highlighting the complex behavior of the generation process in diffusion models.

**Claims And Evidence:**

## Claim

(1) For two-parameter systems, like the classic Ising model and TASEP system, the work introduces metric which reconstructs Fisher metric outperforms existing baselines designed for reconstructing thermodynamics quantities.

(2) The introduced method when applied in diffusion models demonstrate fractal structure of phase transitions in latent space, illustrated by abrupt changes in the reconstructed Fisher metric.

(3) The interpolation based on geodesics' information illustrates a smoother interpolation in contrast to the conventional linear interpolation. Moreover, the authors also claim that the diffusion model exhibits a divergent Lipschitz constant with respect to the latent space at phase boundaries, using geodesics' information as part of their analysis.


## Evidence

(1) To support the first claim, the work first show that the posterior $p(t | x)$ satisfies
$\underset{N \rightarrow \infty}{\text{lim}} \big ( p(t | x_1, \dots x_N) \big )^{1 / N} \overset{a.s.}{=} \mathrm{e}^{-D_{\log Z(t)} (t, t')}$ where $D_{\log Z(t)} (t, t') = \log Z(t) - \log Z(t') - \langle \nabla_t' \log Z(t'), \, t - t' \rangle$ is the Bregman Divergence. This full proof is detailed in Appendix A. There are some assumptions made, but the main assumption is that the general distribution belongs to the exponential family.

Then, with the posterior $p(t | x_1, \dots x_N)$, the log-partition function $log Z(t)$ can be estimated with the trained parameter $\theta$ using the following loss

$
    \mathcal{L}(\theta) = \int_S \mathrm{D}_\mathrm{JS} \big ( p(t | x_1, \dots x_N) \,\, || \,\,\, p_{\log Z_\theta}  (t | t') \big ) dt'
$

Using $\mathcal{L}$, they computed the free energy of the Ising model and TASEP respectively. They recorded the free energy RMSE (root mean squared error) in Table (1), as well as the RMSE of the partial derivatives w.r.t the function $F$ or the Hamiltonian (total energy). Based on Table (1), it's clear that the work's method performs better since it has much lower RMSE than its baselines.

The accuracy of the computed free energy is verified with the depiction of the actual free energy of the two simple models, see figures (3 and 4).


(2) To tackle the second claim, the authors rely on the usage of a feature extractor $\mathcal{E}$ (in this case CLIP) to approximate the posterior $p(t | x_1, \dots x_N) \approx \mathrm{e}^{- \frac{N}{2} \lVert \mathcal{E}(x) - \mathcal{E}(x') \rVert^2}$, where $x \sim p(x|t)$ and $x' \sim p(x | t')$, for the high-dimensional (or image) setting.

Using the loss function $\mathcal{L}$ (mentioned above), the authors trained an MLP which represents $g_F (t) = \nabla^2 \log Z_{\theta^*} (t)$ obtained via $\theta^* = \text{argmin}_\theta \mathcal{L} (\theta)$. Using this network, they are able to observe the phase transitions detailed in figure (7). Keep in mind, they represent t as two interpolation parameters, $\alpha$ and $\beta$.

(3) With the fisher metric $g_F(t) = \nabla^2 \log Z_{\theta^*} (t)$, the method is extrapolated to explore the geodesics' geometry, or shortest path geometry. To do this, the authors rely on the work of [Shao et al. (2018)](https://openaccess.thecvf.com/content_cvpr_2018_workshops/papers/w10/Shao_The_Riemannian_Geometry_CVPR_2018_paper.pdf) for its approach to find the geodesic along a perturbation/interpolation trajectory of two images, by minimizing the curve length $L[\gamma(t), g_f(t)]$ (see Eq. 25).

Although the results are mostly qualitative, we can see that in figures (6 and 7) as well as those in the Appendix, figures (11 and 12) --- that the geodesics' interpolation obtained via minimizing $L$ is much smoother.

## Strength

(1) This is a very interesting paper which presents a method on obtaining information from the (intractable) partition function of diffusion model. Through some formulations, the partition function (based on t), which the work uses to analyze the behavior of diffusion model, is shown to be very informative. Moreover, the results are very interesting.

(2) The formulations/derivations are rigorous and based on or inspired heavily by works done by [Amari and Armstrong (2013)](http://arxiv.org/pdf/1312.1103) and [Bryant (2024)](https://arxiv.org/abs/2405.06998). Hence, I think the paper has a nice story in which it follows very well. The presentation is good.


## Weakness

(1) With the exception of Table (1), in general the results do feel more qualitative than it is quantitative. Also, the results in Table (1) indicate that for the two simple models. It would be nice to have something similar for comparing geosidics and linear interpolation. Perhaps, something like Table (2) of [Shao et al. (2018)](https://openaccess.thecvf.com/content_cvpr_2018_workshops/papers/w10/Shao_The_Riemannian_Geometry_CVPR_2018_paper.pdf).

(2) For the diffusion model analyses, the experiment was conducted only on a stable diffusion model. It would be nice to see this behavior for non-stable diffusion models as well.

**Essential References Not Discussed:**

Since you mentioned Ising model, I think it's also fair to cite Hopfield models in the work as well for relevancy. See [Amari, s-i. (1972)](https://ieeexplore.ieee.org/document/1672070) and [Hopfield J. (1982)](https://www.pnas.org/doi/10.1073/pnas.79.8.2554)

Another interesting work which relies on the analysis of the Jacobian of the score or Hessian of the energy is [Achilli et al. (2024)](https://arxiv.org/pdf/2410.08727?).

**Experimental Designs Or Analyses:**

Yes, I did. Please see what I've said above. Nonetheless, I do not see problems with their experimentation, but rather I think their mathematical works can be written and organized slightly better.

**Methods And Evaluation Criteria:**

In general, I believe the evaluation makes sense. Unfortunately, there are no baselines for the proposed method in high dimensional, except for the 2D model case (since you can compute the actual free energy for Ising and TASEP). It is rather difficult to have actual baselines (to my knowledge).

**Other Comments Or Suggestions:**

Overall, I believe this is a good work, but it needs an additional quantitative result. For example, something like Table (2) of [Shao et al. (2018)](https://openaccess.thecvf.com/content_cvpr_2018_workshops/papers/w10/Shao_The_Riemannian_Geometry_CVPR_2018_paper.pdf) is probably sufficient.

Moreover, there are minor notation errors that should be correct --- and the semantics of some variables (especially those in the Appendix) should be included to make it clear to the readers.

In the main text:

(1) Eq. (22) is labeled as $\mathcal{L}_1$ but shouldn't it just be $\mathcal{L}$ instead? I don't see this subscript 1 being used at all.

(2) In section 4.1, on line 304-305, you should use $\partial$ instead of $\mathrm{d}$ to indicate partial derivative.

In the Appendix:

(1) On line 617, it should be "where $\psi(s, x)$ ..." instead of "where $\psi(x, s)$ ..." just for consistency.

(2) What exactly does the variable $\mathbf{N}$ mean on lines 720 and 753? Do you mean to use $\mathbb{N}$ instead?

Finally, here is my ***main suggestion***. I think you should write an algorithm section detailing how you train your model $g_F(\theta, t)$. It should include the usage of the feature/encoder model. Furthermore, please include the details of the algorithm found in [Shao et al. (2018)](https://openaccess.thecvf.com/content_cvpr_2018_workshops/papers/w10/Shao_The_Riemannian_Geometry_CVPR_2018_paper.pdf) which you relied on, in the Appendix. It will help keep things clear for the readers.

**Other Strengths And Weaknesses:**

See Claim and Evidence Section

**Questions For Authors:**

See Above Section

**Relation To Broader Scientific Literature:**

I think this work is pretty important. We know that diffusion models are just energy-based models (EBMs), which learns the score or the gradient of the energy function (or log-p). Unlike normal EBMs, to compute the log-likelihood or an estimate to the energy, we have to perform a numerical integration of the probability flow ODE, detailed in [Song et al. (2021)](https://arxiv.org/abs/2011.13456). This is simply a way of telling how good a diffusion model learns. But it doesn't fully tell us how diffusion models behave.

This work presents an interesting approach and a point of view of how we can interpret the intractable partition function, which we avoided in training diffusion models, such that we can analyze the behavior of diffusion models.

**Theoretical Claims:**

The paper definitely makes a lot of theoretical claims. Thus, relying heavily on works done by others, the authors were able to rigorously formulate their approach and also proofs behind it. See Appendix A for the most important proof.

---

> ### Author Rebuttal · Authors · 2025-04-01
>
> We highly appreciate provided feedback and agree that the paper will benefit from additional quantitative studies.
>
> - Providing additional quantitative result, like Table (2) of Shao et al. (2018)
>
>   Please refer to the General Response below.
>
> - Adding experiments for non-stable diffusion generative models
>
>   In addition to stable diffusion, we have applied the proposed approach to StyleGAN3 (2021). However, it resulted in geodesics close to linear interpolation and overall our findings coincide with Wang & Ponce (2021), so these experiments are not included in the main paper text. Please find Figure 9: "Free energy surface of a StyleGAN v3 reconstructed using CLIP distance" in supplementary material.
>
> - Writing an algorithm section detailing how the model is trained, including the details of the algorithm found in Shao et al. (2018)
>
>   We agree that these algorithmic details should be written explicitly.
>
> # The General Response
> We would like to thank the reviewers for recognizing our method to approximate Fisher information metric as novel and theoretically grounded.
> Since most reviews raise the question of comparing our work with Wang & Ponce (2021) and Shao et al. (2018), we would like to provide a detailed comment on the distinction between our proposed method and these prior works.
>
> Shao et al. studies VAE and computes geodesic curves based on pullback metric induced by euclidean metric in pixel space. Wang et al. consider GAN models and defines metric on the latent space via pullback of L2 distance in the feature space of VGG-19 (LPIPS distance). In both cases generative models are deterministic: each latent Z produces only a single image X.
>
> The key distinction of our work lies in its consideration of a broader class of models, specifically, models with stochastic generation, where a single latent Z corresponds to a distribution in the image space p(X|Z). Diffusion models with stochastic sampling (and all statistical physics models) cannot be addressed within the formalism of Shao et al. (2018) or Wang et al. (2021).
>
> To fulfill the request of the reviewers we compare our algorithm with Shao et al. (2018) and Wang et al. (2021) in deterministic sampling regime. We get pullback metric from the euclidean metric in the space of CLIP embeddings. The Jacobian is estimated via finite differences. We use three evaluation scores: CLIP, pixel and Perceptual Path Length (PPL), which measure the average path length in CLIP embedding space, pixel space and feature space of VGG-19. We compute them as the cumulative L2 distance between consecutive feature vectors (or images for pixel space), then average this perceptual divergence measure across multiple trajectories.
>
>     Average CLIP Lengths ± std (bootstrap):
>     --------------------------------------------------
>     Geodesic ours: 72.2951 ± 3.7516
>     Linear:        73.6076 ± 3.5428
>     Geodesic:      73.5764 ± 4.3664
>
>
>     Average Pixel Lengths ± std (bootstrap):
>     --------------------------------------------------
>     Geodesic ours: 2769146.3818 ± 23813.2801
>     Linear:        2757206.0770 ± 27731.2199
>     Geodesic:      2739432.4217 ± 35251.5666
>
>
>     Average Perceptual Path Lengths ± std (bootstrap):
>     --------------------------------------------------
>     Geodesic ours: 3.1172 ± 0.1587
>     Geodesic:      3.1857 ± 0.2134
>     Linear:        3.1725 ± 0.2256
>
>
> We observe that our interpolation is on par with Shao et al. (2018) and Wang et al. (2021) in the case of deterministic sampling. We additionally compute the curvature of each trajectory as the mean angular change per unit length between consecutive path segments. Our analysis reveals that trajectories constructed using the Wang metric show significantly higher curvature and frequent turning compared to our method. We attribute this behavior to the finite differences, which introduce high frequency noise in metric components. In the case of diffusion, the Jacobian is hard to obtain via backpropagation as suggested in Shao et al. (2018) due to high computational cost.
>
>     Average Mean Curvature ± std (bootstrap):
>     --------------------------------------------------
>     Geodesic ours: 0.3671 ± 0.6909
>     Geodesic:      1.3261 ± 0.5328
>     Linear:        0.0000 ± 0.0000

---

> > ### Comment · Reviewer_GBhG · 2025-04-02
> >
> > Dear authors,
> >
> > Thank you for your response and hard work on your new findings. In my opinions, I believe these new findings are quite interesting. However, I dislike the fact that you did not denote whether these values are supposed to be good or bad depending on whether they are higher or lower. This will most likely confuse other reviewers. Anyway, I do have my interpretation about your new results --- I think your results on the mean curvature and avg. CLIP lengths and avg. Perceptual Path lengths are good.
> >
> > I understand there are still some issues with other reviewers to be discussed. But I think the works which this paper follows are quite rigorous and provide some good background and motivation. The approach, presented in this paper, is certainly not adaptable and scalable to higher dimension (if I recall from reading correctly), but it is a different way to interpret the problem of analyzing un-normalized generative models. I find it quite fascinating, frankly.
> >
> > Nonetheless, I would like to keep my score and I wish the authors best of luck.

---

### Official Review · Reviewer_wCzF · 2025-03-10

**Overall Recommendation:** 1

**Summary:**

This paper proposes a novel method to approximate the Fisher information metric. It shows superior performance to concurrent methods on Ising and Tasep models. Applied to diffusion models, it reveals a fractal structure of the latent space, and sharp transitions. It also allow for smoother interpolations.

## Update

I will keep my rating as is, since I won't be able to evaluate the paper revision and the newly added presentation/details. Moreover, experimental results are still not particularly strong and convincing when considering std of metrics.

**Claims And Evidence:**

The paper claims that their method allows a good approximation of the Fisher information metric, which allows its use in thermodynamic models and on analysing latent space of diffusion models. To me, the empirical evidence is not strong enough (see below).

**Essential References Not Discussed:**

Not essential but would be worth discussing: 1: "Metrics for Deep Generative Models", Chen et al., AIStats 2018.
2: "Optimal transport maps for distribution preserving operations on latent spaces of Generative Models.", Agustsson et al., 2018.

**Experimental Designs Or Analyses:**

As mentioned above, the evaluation criteria is weak in my opinion. The first part is on toy data. The second part lacks comparisons with baselines.

**Methods And Evaluation Criteria:**

Method: The method is sound and well thought.

Evaluation criteria: The first evaluation criteria is a validation on exact statistical models for thermodynamics. I do not know well these models, and thus I am unsure of the interest of these results.
The second evaluation criteria is on the analysis of the latent space of diffusion models. The main weakness is the lack of comparisons with other methods. Why does the proposed method reveal things that would not be revealed by other methods? For example, it would be worth adding a simple baseline measuring the generator's local curvature (e.g. estimated with finite difference), which could also reveal these phase transitions. Moreover, it lacks metrics. Validating an idea based on a few visualizations is not rigorous enough. A potential metric could be the Perceptual Path Length introduced in StyleGan paper.

**Other Comments Or Suggestions:**

No.

**Other Strengths And Weaknesses:**

Additionally to the lack of strong experiments, a second weakness is the paper clarity and the low level of details. To me, the method is not detailed enough. For example, what is described in 3.1 and 3.2 should be summarized in an Algorithm. Moreover, details required to reproduce the experiments are lacking, which is a problem.

**Questions For Authors:**

No.

**Relation To Broader Scientific Literature:**

The method described here is a novel method to approximate Fisher information metric in my knowledge.

**Theoretical Claims:**

I did not check the proofs. However, let us note that the theory proves the feasibility of their method rather than its superiority compared to other methods.

---

> ### Author Rebuttal · Authors · 2025-04-01
>
> We thank the reviewer for highlighting gaps in our evaluation and presentation. The revision will strengthen empirical validation with baselines and metrics and improve clarity via algorithms and reproducibility details. To address concerns about comparison with other methods please refer to tables below. We hope these changes will address the concerns and demonstrate the method’s utility for analyzing generative model geometries.
> # The General Response
> We would like to thank the reviewers for recognizing our method to approximate Fisher information metric as novel and theoretically grounded.
> Since most reviews raise the question of comparing our work with Wang & Ponce (2021) and Shao et al. (2018), we would like to provide a detailed comment on the distinction between our proposed method and these prior works.
>
> Shao et al. studies VAE and computes geodesic curves based on pullback metric induced by euclidean metric in pixel space. Wang et al. consider GAN models and defines metric on the latent space via pullback of L2 distance in the feature space of VGG-19 (LPIPS distance). In both cases generative models are deterministic: each latent Z produces only a single image X.
>
> The key distinction of our work lies in its consideration of a broader class of models, specifically, models with stochastic generation, where a single latent Z corresponds to a distribution in the image space p(X|Z). Diffusion models with stochastic sampling (and all statistical physics models) cannot be addressed within the formalism of Shao et al. (2018) or Wang et al. (2021).
>
> To fulfill the request of the reviewers we compare our algorithm with Shao et al. (2018) and Wang et al. (2021) in deterministic sampling regime. We get pullback metric from the euclidean metric in the space of CLIP embeddings. The Jacobian is estimated via finite differences. We use three evaluation scores: CLIP, pixel and Perceptual Path Length (PPL), which measure the average path length in CLIP embedding space, pixel space and feature space of VGG-19. We compute them as the cumulative L2 distance between consecutive feature vectors (or images for pixel space), then average this perceptual divergence measure across multiple trajectories.
>
>     Average CLIP Lengths ± std (bootstrap):
>     --------------------------------------------------
>     Geodesic ours: 72.2951 ± 3.7516
>     Linear:        73.6076 ± 3.5428
>     Geodesic:      73.5764 ± 4.3664
>
>
>     Average Pixel Lengths ± std (bootstrap):
>     --------------------------------------------------
>     Geodesic ours: 2769146.3818 ± 23813.2801
>     Linear:        2757206.0770 ± 27731.2199
>     Geodesic:      2739432.4217 ± 35251.5666
>
>
>     Average Perceptual Path Lengths ± std (bootstrap):
>     --------------------------------------------------
>     Geodesic ours: 3.1172 ± 0.1587
>     Geodesic:      3.1857 ± 0.2134
>     Linear:        3.1725 ± 0.2256
>
>
> We observe that our interpolation is on par with Shao et al. (2018) and Wang et al. (2021) in the case of deterministic sampling. We additionally compute the curvature of each trajectory as the mean angular change per unit length between consecutive path segments. Our analysis reveals that trajectories constructed using the Wang metric show significantly higher curvature and frequent turning compared to our method. We attribute this behavior to the finite differences, which introduce high frequency noise in metric components. In the case of diffusion, the Jacobian is hard to obtain via backpropagation as suggested in Shao et al. (2018) due to high computational cost.
>
>     Average Mean Curvature ± std (bootstrap):
>     --------------------------------------------------
>     Geodesic ours: 0.3671 ± 0.6909
>     Geodesic:      1.3261 ± 0.5328
>     Linear:        0.0000 ± 0.0000

---

### Official Review · Reviewer_Evof · 2025-03-13

**Overall Recommendation:** 3

**Summary:**

This paper leveraged the information geometry framework to understand the manifold geometry in statistical mechanics models parameter space and generative model latent space.

They provided nice theoretical results connecting the posterior distribution of parameter given infinite samples and the Bregman divergence of the parameter and the true ones. Further they tried to use this connection to learn the partition function Z, whose Hessian gives us the Fisher metric of the manifold. Then they use neural networks to estimate posterior distributions given samples and to learn the log partition function. They validated the pipeline on some classic statistical physcis model showing it can learn the free energy better than baseline. They also used it to study the latent space geometry in diffusion models, which elucidates a fractal structure in latent space.

## Update after rebuttal
The authors clarified most conceptual concerns regarding the method. (esp. the concerns about latent vector falling off shell in the final interpolation experiment.) The reviewer is happy to remain their score.

**Claims And Evidence:**

- In the paper it sometimes said colloquially, “*estimating the log partition function log Z(t) by training a network to simulate p(t|x)*” But we can only know the log Z up to affine transform. So maybe we can qualify the claim about estimating log Z(t) ?
- The formulation makes a lot of sense for statistical mechanics setting, and I’m happy to see it learning the correct partition function.
But for the generative image model latent space setting (with feature extractor), it seems a bit overly complex. I’m not clear / convinced that this framework is better / more informative than the previous ones to compute the Riemannian metrics for the latent space of generative models (see related works [1,2]). So I don’t buy the claim that “*Notably, the approach proposed below is theoretically justified for any generative model*”

[1] Shao, H., Kumar, A., & Thomas Fletcher, P. (2018). The riemannian geometry of deep generative models.

[2] Wang, B., & Ponce, C. R. (2021). The geometry of deep generative image models and its applications. ICLR

**Essential References Not Discussed:**

In the introduction section “*Riemannian geometry of latent space*” the authors forgot to cite / discuss one important paper along this line [^2], which also managed to compute and analyzed the structure of a Riemannian (Hessian) metric for the latent space of GANs. Since GAN is a deterministic mapping from latent space to image space, their Hessian metric of latent space is derived by pulling back a image distance function (e.g. LPIPS) and compute its Hessian at each point. Thus, in their case the potential function is $\phi_x(x’)=D(G(x),G(x’))$. The potential function is local to each point, and not necessarily global. But that framework can deal with higher dimensional manifolds > 2d, though without the gurantee as in *(Bryant–Amari–Armstrong)*.

The overall conceptual framework of the current paper share many similarities with [3], so it would be nice to discuss the connection and shared spirit with them.

[2] : Wang, B., & Ponce, C. R. (2021). The geometry of deep generative image models and its applications, ICLR, 2021

**Experimental Designs Or Analyses:**

- Experimental design and results in Fig. 3-4 are convincing, showing that the inference method can recover known results.

Questions:

- **Major point: Random selection of 3 initial latents and chart the 2d plane between them.**
I feel this design has the risk of falling off the manifold of Gaussian hypersphere. Since the trained latents were sampled from N(0,I), as long as your z_0,z_1,z_2 are not too close, many of their interpolations will be “off” the distribution and have incorrect norm, thus I’m not sure the score function / the sampler can do the right thing for those samples.
Can you consider a 2d submanifold on the sphere, similar to the way people interpolate latent space in GAN era ([3] [4] Fig1E).
    - In previous works, people seems to find the manifold can look quite continuous if we perturb the initial state in the “good directions” found by PC (see [5] Fig.5 and supp). So I’m not sure if the fractal structure you found in your Fig. 6 is because we are charting a subspace in latent space that were unnatural for diffusion models.
- **Minor point:** For Fig. 6, it’s very cool to show this fractal boundary perceptually, is there a way to quantify the “fractalness” of it?
- **Minor point:** For Fig. 7C, we see pretty linear border separating the “phase” on the 2d space. I feel part of it is due to the model you used to approximate $\log Z(t)$ is a MLP with ReLU, which is piecewise linear. Thus their separating boundary have to be piecewise linear in a sense.

[3] White, T. (2016). Sampling generative networks. arXiv preprint arXiv:1609.04468.

[4] Wang, B., & Ponce, C. R. (2022). Tuning landscapes of the ventral stream. *Cell Reports*, *41*(6).

[5] Wang, B., & Vastola, J. J. (2023). Diffusion models generate images like painters: an analytical theory of outline first, details later. *arXiv:2303.02490*.

**Methods And Evaluation Criteria:**

Yes. Use of solvable statistical physics models as a test case suits the theoretical framework of the authors.

**Question:**

- **Major point**
The Eq 16-19 is a nice theoretical formulation for using feature extractor.
But in the end seems Eq.19 is not very differently from the previous Jacobian formulations, i.e. computing squared distances in some feature space and pulling back to the latent space. [2] In the end if we are using feature extractors, how does the metric you got for diffusion latent manifold similar / different from the metric obtained from pulling back the euclidean metric in feature space?
That could be done by simple finite difference method if you have all the sample images, no need to backprop through the diffusion sampling process. I guess it may not be more expensive than the MLP training in your cae.
- When you are using feature extractors, why should we learn log Z(t) ? I feel learning a a bivariable distance function d(t, t’) could also be possible or even faster to approximate the D_log Z(t,t’) directly.
- **Minor point:** In methods Sec 3.1, the neural network design were quite suitable for 2d settings, i.e. the  Ising models are 2d and the parameter space is also 2d, so you can do 2d→2d unet mapping. But seems it’s hard to generalize this to higher dimensional parameter space?

[2] Wang, B., & Ponce, C. R. (2021). The geometry of deep generative image models and its applications. ICLR

**Other Comments Or Suggestions:**

No.

**Other Strengths And Weaknesses:**

- The authors put efforts into the formalism and well made illustrations; they helped understanding a lot.
- The connection the authors draw between generative models and statistical physics models are interesting and quite original.
- Fig. 6 the visualization of the fractal and self-similar structure of the phase boundary within diffusion model latent space is quite novel and intriguing!

**Questions For Authors:**

See above.

**Relation To Broader Scientific Literature:**

- The multiple links drawn between statistical physics and generative models were intriguing, different from many other previous links in DL theory.

**Theoretical Claims:**

I skim through proof of Theorem 3.1 and Lemma 3.3 which seems to be correct. I didn’t check theorem 3.2

**Question:**

- **Major point:**
I cannot quite follow the logic from Eq.21 to 24. esp. 22. Why we can learn *log Zθ(t)* by minimizing the JS divergence in Eq.22? Is it because Theorem 3.1? Do we have some theoretical properties about the objective in Eq.22? Also does Eq. 21 require the latent space you are studying is compact, or the integration is impossible?
- **Minor point:**
from Eq. 16-19 the formalism is slightly off, I feel it’s mixing the deterministic sampling case and the stochastic sampling case…. esp. Eq. 19
- **Minor point:** In the degenerate case, where the generative mapping is deterministic, or $p(x|t)$ is delta measure, does the theory still work? This is relevant since for diffusion, if sampler is an ODE solver, then the mapping is deterministic. Similarly for latent space of GAN. I feel authors need to show this to justify the claim “*approach proposed below is theoretically justified for any generative model*”
    - My hunch is that it (Eq. 4) will fall back to the Jacobian metric, which is the pull back metric of Euclidean metric in x space. Then it will be quite similar to the metric studied by previous works [0][3].

[1] Shao, H., Kumar, A., & Thomas Fletcher, P. (2018). The riemannian geometry of deep generative models.

[2] Wang, B., & Ponce, C. R. (2021). The geometry of deep generative image models and its applications. ICLR

---

> ### Author Rebuttal · Authors · 2025-04-01
>
> We thank author for the elaborated questions regarding theory and method.
>
> - The proposed approach seems a bit overly complex, not clear whether it is better than established approach, that is, pulling back the euclidean metric in feature space [Wang & Ponce 2021] [Shao, Kumar, & Fletcher, 2018].
>
>   We admit that the proposed algorithm is more complex than Wang et al. (2021) and Shao et al (2018). However, we would like to highlight that this complexity comes from the broader coverage of the algorithm, namely, its ability to work with stochastics sampling, where pulling back the euclidean metric in feature space is impossible and results in stochastic metric tensor.
>
>
> - Eq.19 is not very different from the previous Jacobian formulations
>
>   For numerical comparison with previous work please refer to the General Response section in responses to reviewers wCzF and GBhG for details.
>
> - The transition from Eq.21 to 24. esp. 22. Why we can learn $log Z_θ(t)$ by minimizing the JS divergence in Eq.22? Do we have some theoretical properties about the objective in Eq.22?
>
>   Regarding the eq.22, in general one could use other loss functions for distance between distributions. We discuss after Theorem 3.1 that MSE loss exhibits vanishing gradients and thus is not suitable in our case. Since Jensen-Shannon divergence is the proper metric on probability distributions therefore its convergence to zero guarantees the convergence of MSE loss.
>
> - Also does Eq. 21 require the latent space you are studying is compact, or the integration is impossible?
>
>   Yes, latent space has to be compact and we require that integration over it is possible.
>
> - If the generative mapping is deterministic (in a case delta measure), does the theory still work?
>
>     Since delta measure could be uniformly approximated by Gaussians and for any Gaussian the theory is justified, it can be proven the approach still works in the limit.
>
> - This design has the risk of falling off the manifold of Gaussian hypersphere
>
>   Indeed, this is a valid point which is missed in the main text. To prevent our interpolation to fall off the Gaussian hypersphere in all our experiments we employed a normalization of latent vector z.

---

> > ### Comment · Reviewer_Evof · 2025-04-07
> >
> > Thank you for the concise response. The falling-off manifold & normalization point is crucial for evaluating the results, and the authors should emphasize it in the final version of the paper.
> > We will keep the score as is.

---

### Official Review · Reviewer_ACqo · 2025-03-14

**Overall Recommendation:** 2

**Summary:**

This paper introduces a novel approach to exploring the geometric structure of latent
spaces in generative models by estimating the Fisher metric and investigating phase
transitions. Through theoretical developments and empirical validation on statistical
physics models and Stable Diffusion. Key findings include the identification of
distinct phases within the latent space, fractal-like boundaries between these phases,
and the property that geodesic interpolations are linear within phases while exhibiting
nonlinear behavior at phase boundaries. Overall, this work provides new insights into
the latent space of generative models.

**Claims And Evidence:**

Claims : A novel method to reconstruct the Fisher metric in latent spaces of
generative models.

Evidence 1: The authors give theoretical foundations provided through Theorems
3.1 and 3.2.

**Essential References Not Discussed:**

A brand new information geometry method and I cannot give relative references.

**Experimental Designs Or Analyses:**

Experiment primarily validated on specific models(Ising and TASEP) and stable
diffusion models. But there exist Generalizability Concerns. The experiment primarily
validated on specific models but it is unclear universal applicability across all
generative models.

**Methods And Evaluation Criteria:**

Using u2-net and CLIP for different generative models. The authors compare the
RMSE in the statistical physical models experiment and did geodesic
interpolation analysis and phase transition identification in diffusion models
experiment.

Problems : When doing the feature extraction, different feature extractors could
yield varying results. What may happen if the authors use different feature
extractors.

**Other Comments Or Suggestions:**

none

**Other Strengths And Weaknesses:**

(1) Strengths:

Theoretical Innovation: Bridges multiple disciplines: statistical physics,
information geometry, machine learning

Methodological Contributions: Novel approach to reconstructing Fisher metric
and give theoretical convergence guarantees

(2) Weakness:

Generalizability Limitations: what about other feature extractors and generative
models ?

Computational Complexity: Theoretical proofs suggest scaling challenges and
there may potential computational overhead for large-scale models

**Questions For Authors:**

1) Generalizability: How confident are you that the proposed Fisher metric
reconstruction method can be generalized beyond the specific models and about
different generative models
2) Feature Extraction Sensitivity: How robust is your method to different feature
extraction techniques
3) Computational Complexity: could you addressing scalability challenges in high-
dimensional latent spaces?
4) Modeling application: how could apply this analysis theory and improve the
model performance ?
5) Phase Transition: observed fractal-like boundaries in latent spaces? What underlying physical or
mathematical principles explain these transitions?

**Relation To Broader Scientific Literature:**

none

**Theoretical Claims:**

The research extends our understanding of generative models beyond traditional
perspectives, revealing their intricate geometric structures and phase transition
characteristics. Mathematical details are correct, the proof process is rigorous, and the
use of symbols is standardized. The paper is very solid in theoretical derivation.

---

> ### Author Rebuttal · Authors · 2025-04-01
>
> We thank the reviewer for the feedback and hope that the response below will adress the raised points.
>
> - How confident are you that the proposed Fisher metric reconstruction method can be generalized beyond the specific models?
>
>     We are fairly confident it can generalize to other types of generative models, since it is justified by Bryant–Amari–Armstrong theorem for 2D section of a latent spaces. For higher-dimensional spaces it is not guaranteed to work. Beyond stable diffusion models, we tested the method with StyleGAN3, please see Fig.9: "Free energy surface of a StyleGAN v3 reconstructed using CLIP distance" in supplementary material. The learned surface has a little curvature and exhibits a single phase, which is in agreement with previously done research by Wang & Ponce (2021).
>
> - Different feature extractors could yield varying results
>
>     During our evalutation done with CLIP, pixel-wise distances and PPL (perceprual path length) we obtained grid approximations of local metric following Wang & Ponce (2021). We observed that phase boundaries are stable under tested feature extractors. Because of this, we conclude that our algorithm will learn the same phase boundaries. Please see the General Response section in reponses to reviewers wCzF and GBhG for details.
>
> - Scalability challenges in high-dimensional latent spaces
>
>   Our current method is justified for 2D section of high-dimensional latent spaces as we follow Amari and Armstrong (2013) and Bryant (2024).
>   Though the training procedure can be extended by CNN that predicts parameters of gaussian mixture models, the theoretical ground is the subject of futher research.
>
> - Phase Transition: What underlying physical or mathematical principles explain these transitions?
>
>     We suppose that this phenomenon is related to the structure of image space, as studied in the ICLR 2023 paper “Verifying the Union of Manifolds Hypothesis for Image Data” by Brown et al. In this paper, it is shown that image data lies on a disjoint union of lower-dimensional manifolds with varying dimensions. The generative process of diffusion models begins with a Gaussian distribution defined on a full-dimensional latent space, and the reverse ODE trajectories end at significantly lower-dimensional, disjoint manifolds that represent real image data. Such a mapping—from a higher-dimensional unimodal distribution to a multimodal data manifold with disjoint supports for each mode—may exhibit a diverging Lyapunov exponent, or, in other words, a diverging Lipschitz constant for the generative mapping from the latent space to the data space.
>
>
> This phenomenon can be illustrated by the following Proposition, which simulates a lower-dimensional data manifold with disjoint supports.
>
> # Proposition
> Suppose that the (target) data distribution is a bimodal mixture of two gaussians, each with variance $\sigma$:
> $$
> p_0(x) = \frac{1}{2}\mathcal{N}(x\mid -1,\sigma^2) + \frac{1}{2}\mathcal{N}(x\mid 1,\sigma^2)
> $$
> The latent (source) distribution is the standard normal $\mathcal{N}(x\mid 0,1)$. Consider the Variance Preserving SDE
> $$
> dX_t = -\frac{1}{2}\beta X_t dt + \sqrt{\beta} dW_t
> $$
> Then the Lyapunov exponent of the corresponding reverse-time ODE at $x=0$ has the following form
> $$
> \lambda=\frac{\beta}{2}\left(1+\frac{1-\sigma^2}{\sigma^4}\right)
> $$
> and it diverges to infinity as $\sigma$ goes to zero. In this case the point $x=0$ could be interpreted as a phase transition boundary.
>
> # Proof
> We begin with the reverse probability flow ODE written as
> $$
> \frac{dX_s}{ds}=-f(X_s,t)+\frac{g(t)^2}{2}\nabla_x\log p_t(X_s),
> $$
> where the drift term is
> $$
> v(x)= -f(x,t)+\frac{g(t)^2}{2}\nabla_x\log p_t(x).
> $$
> Linearizing around $x=0$, the Lyapunov exponent is defined by
> $$
> \lambda=v'(0)=-f'(0,t)+\frac{g(t)^2}{2}\frac{d^2}{dx^2}\log p_t(x).
> $$
> The density p_t(x) is computed via convolution with the gaussian noising kernel
> $$
> p_t(x)=\int_{-\infty}^{\infty} p_0(y)\mathcal{N}\Bigl(x\Big|e^{-\frac{1}{2}\beta t}y,1-e^{-\beta t}\Bigr)dy.
> $$
> Since convolution of gaussians is still gaussian we obtain
> $$
> p_t(x)=\frac{1}{2\sqrt{2\pi}\sigma_{1}(t)}A(x)
> $$
> where
> $$
> A(x)=e^{-\frac{(x-\mu(t))^2}{2\sigma_1^2(t)}}+e^{-\frac{(x+\mu(t))^2}{2\sigma_1^2(t)}}
> $$
> and
> $$
> \sigma_{1}^2(t)=e^{-\beta t}\sigma^2+(1-e^{-\beta t})
> $$
> Then by computing derivatives in the definition of Lyapunov exponent we get
> $$
> \lambda = \frac{\beta}{2} + \frac{\beta}{2} \frac{e^{-\beta t} - \sigma_1^2(t)}{\sigma_1^4(t)}
> = \frac{\beta}{2} (1 + \frac{e^{-\beta t} - \sigma_1^2(t)}{\sigma_1^4(t)})
> $$
> As time goes to 0 we get $\sigma_1 \rightarrow \sigma$, and have diverging Lyapunov exponent for small $\sigma$
> $$
> \lambda=\frac{\beta}{2}\left(1+\frac{1-\sigma^2}{\sigma^4}\right)
> $$
> suggesting divergence of close reverse ODE trajectories.
>
> End of the Proof.

---

### Decision · Program_Chairs · 2025-05-01

**Decision:**

Accept (poster)

**Comment:**

There is an even split between the scores of the reviewer, but sadly there was no discussion whatsoever on the topic of reaching a consensus. On the positive side, there was a healthy interaction between the reviewers and the authors post-rebuttal.

I am convinced regarding the novelty and the experimental evaluation of the proposed method.
I would strongly suggest the authors take the time to update the manuscript so it incorporates the reviewers' feedback. I believe the additional results around the Avg mean curvature belong in the main paper and I would like to see a nice discussion around them.

Congratulations!